# Learning Energy-Based Models by Self-Normalising the Likelihood

**Hugo Henri Joseph Senetaire**                                    *hhjs@dtu.dk*
*Technical University of Denmark*

**Paul Jeha**                                                      *pauje@dtu.dk*
*Technical University of Denmark*

**Jes Frellsen**$^*$                                               *jefr@dtu.dk*
*Technical University of Denmark*

**Pierre-Alexandre Mattei**$^*$              *pierre-alexandre.mattei@inria.fr*
*Université Côte d'Azur*
*Inria, CNRS*
*LJAD, Maasai team*

**Reviewed on OpenReview:** *https: // openreview. net/ forum? id= GVaPBqI6ny*

## Abstract

Training an energy-based model (EBM) with maximum likelihood is challenging due to the intractable normalisation constant. Traditional methods rely on expensive Markov chain Monte Carlo (MCMC) sampling to estimate the gradient of logartihm of the normalisation constant. We propose a novel objective called self-normalised log-likelihood (SNL) that introduces a single additional learnable parameter representing the normalisation constant compared to the regular log-likelihood. SNL is a lower bound of the log-likelihood, and its optimum corresponds to both the maximum likelihood estimate of the model parameters and the normalisation constant. We show that the SNL objective is concave in the model parameters for exponential family distributions. Unlike the regular log-likelihood, the SNL can be directly optimised using stochastic gradient techniques by sampling from a crude proposal distribution. We validate the effectiveness of our proposed method on various density estimation and parameter estimation tasks. Our results show that the proposed method, while simpler to implement and tune, outperforms existing techniques on small to moderate dimensions but its performance starts to degrade for very high-dimensional problems. We extend this framework to handle EBM for regression and show the usefulness of our method in this setting as we outperform existing techniques.

## 1 Introduction

Energy-based models (EBMs) specify a probability density over a space $\mathcal{X}$ through a parameterised energy function $E_\theta : \mathcal{X} \to \mathbb{R}$. The associated density is then

$$p_\theta(x) = \frac{e^{-E_\theta(x)}}{Z_\theta}, \tag{1}$$

where $Z_\theta = \int e^{-E_\theta}(x)\,\mathrm{d}x$ is called the partition function or the normalising constant. However, $Z_\theta$ is often unknown and intractable, which makes training an EBM through maximum likelihood challenging.

---

$^*$Equal contribution.

Initial methods address the challenge with a pseudo-likelihood function, an altered version of the likelihood function that circumvents the need to compute the normalising constant (Besag, 1975; Mardia et al., 2009; Varin et al., 2011). Alternatively, gradients of the log-likelihood function can be estimated using the Boltzmann learning rule (Hinton & Sejnowski, 1983) or approximated using contrastive divergence (Hinton, 2002) at the price of expensive and difficult-to-tune Markov chain Monte Carlo (MCMC) sampling methods (Dalalyan, 2017; Welling & Teh, 2011). To alleviate this difficulty, Du & Mordatch (2019) proposed to maintain a buffer of samples during training using Langevin MCMC. This work was extended by Du et al. (2021), who considered a Kullback-Leibler divergence term that was claimed to be negligible by Hinton (2002). Relatedly, Xie et al. (2021) use a flow trained alongside the EBM as a starting point for a short-term MCMC sampler, reducing the dependency on long chains. In another work, Gao et al. (2021) proposed training a succession of EBM on data diffused with noise, allowing for both training and sampling the conditional distribution. Nijkamp et al. (2019) studied the training of EBM for short-term non-convergent Langevin Markov chains and showed excellent generation, albeit without directly optimising the likelihood. As it is critical to have a good estimate of this gradient, alternative methods consider using a proposal distribution $q$ together with

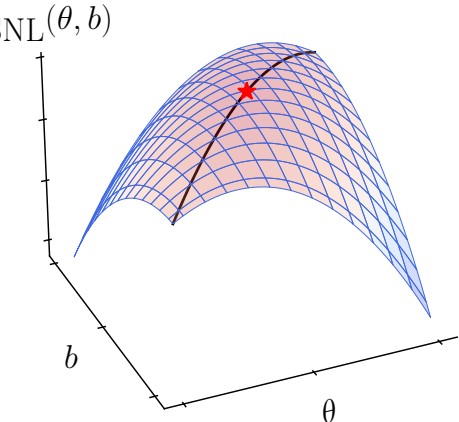

Figure 1: The SNL for a Gaussian with unknown mean $\theta \in \mathbb{R}$ and unit variance. The SNL a function of both $\theta$ and the additional parameter $b$, estimating the normalising constant. The black line corresponds to maximising $b$ for each given $\theta$, which exactly recovers the log-likelihood. The red star is the maximum log-likelihood, which is also the maximum of $\ell_{\mathrm{SNL}}(\theta, b)$, see details in Appendix B.

importance sampling (Bengio & Senécal, 2003). However, this results in an objective that is an upper bound of the log-likelihood. Additionally, the choice of a proposal is critical, and a poor choice will lead to a loose bound. To tighten it, Geng et al. (2021) train the proposal to minimise the bound. This results in a min-max objective, similar to that of generative adversarial networks (GANs), which are infamous for their instability in training (Kumar et al., 2019; Farnia & Ozdaglar, 2021).

Another line of work aims at getting rid of the partition function altogether. It notably includes score matching and its variants (Hyvärinen, 2005; Vincent, 2011). Score matching is a family of objectives that circumvents the normalising constant by matching the Stein score of the data distribution (Stein, 1972) to that of the model. Several variants have subsequently been proposed: implicit score matching trades the Stein score of the data distribution for the Hessian of the model (Hyvärinen, 2005; Kingma & Le Cun, 2010; Martens et al., 2012), while denoising score matching models instead a corrupted version of the data, which has a tractable density. The latter approach has proven very successful at generating high-dimensional data, such as images and videos (Song et al., 2021; Ho et al., 2022). Another approach that bypasses the normalising constant is to minimise the Stein discrepancy (Barp et al., 2019; Grathwohl et al., 2020).

An alternative approach more closely related to our work is noise contrastive estimation (NCE), where Gutmann & Hyvärinen (2010) frames the problem as a logistic regression task between the data and a tractable noise distribution. This leads to a consistent estimate of the model parameters. Additionally, the normalisation constant is learned using an additional parameter (Mnih & Teh, 2012). The crucial issue of NCE, and that will not affect our method, is that the objective depends on the noise distribution, which is very hard to optimise (Chehab et al., 2022).

## 1.1 Contributions

Our work is inspired by two papers on local likelihood density estimation (Loader, 1996; Hjort & Jones, 1996), which mention ways of bypassing the normalising constants in their quite specialised context. Our contributions are the following:

- We propose a new objective, the self-normalised log-likelihood (SNL) that is amenable to stochastic optimisation and allows for recovering both the maximum likelihood estimate and its normalising constant.

- We study theoretical properties of the SNL, in particular its concavity for exponential families and its links with information geometry.

- We demonstrate on a range of low-dimensional tasks, including density estimation and parameter estimation, that SNL is straightforward to implement and achieves performance comparable to that of more complex approaches for learning energy-based models. We show state-of-the-art results on image regression datasets using an energy-based model.

- We derive a surrogate training objective, the SNELBO, for variational autoencoders with an EBM prior, and evaluate it on binary MNIST, CIFAR-10, and CelebA. While this approach improves upon the vanilla VAE baseline, the resulting generations remain below the performance of current state-of-the-art models.

## 2 Self-normalising the likelihood

We deal with some data $x_1, \ldots, x_n \in \mathcal{X}$, assumed to be independent and identically distributed samples from a distribution $p_{\text{data}}$. Our goal is to fit an EBM $p_\theta$, as defined in Eq. (1), to these data. The standard approach for fitting a probabilistic model is to maximise the likelihood function

$$\ell(\theta) = \frac{1}{n} \sum_{i=1}^{n} \log p_\theta(x_i). \tag{2}$$

Unfortunately, as we will discuss now, maximising such a function for an EBM is a daunting task.

### 2.1 Why maximum likelihood for EBMs is hard

Let us focus on a single data point $x$. The log density of our EBM is

$$\log p_\theta(x) = -E_\theta(x) - \log Z_\theta, \tag{3}$$

with $\theta$ being the learnable parameters of the model. Gradient-based methods are a popular approach to train an EBM via maximum likelihood; those methods require the gradient of the log density with respect to the parameters, $\theta$, that is

$$\nabla_\theta \log p_\theta(x) = -\nabla_\theta E_\theta(x) - \nabla_\theta \log Z_\theta. \tag{4}$$

While automatic differentiation can, usually, easily compute the gradient of the energy $\nabla_\theta E_\theta(x)$, it is not the case for $\nabla_\theta \log Z_\theta$. However, following the Boltzmann learning rule (Hinton & Sejnowski, 1983), we can express the gradient of the normalising constant as an expected value (see, e.g., Song & Kingma, 2021 for a full derivation):

$$\nabla_\theta \log Z_\theta = -\mathbb{E}_{X \sim p_\theta}[\nabla_\theta E_\theta(X)]. \tag{5}$$

We can obtain a Monte Carlo estimate of this gradient, but this requires sampling from the EBM itself, which leads to the use of MCMC-based methods that often suffer from poor stability and high computational cost. These procedures usually require very long chains to converge to the true distribution $p_\theta$. For the EBM to be computationally trainable, one needs to cut short the procedure, and as a result, the obtained samples do not follow exactly $p_\theta$, meaning that the estimates of $\nabla_\theta \log Z_\theta$ are biased. As it is critical to have a good and fast estimate of this gradient, alternative methods consider using a proposal distribution $q$ in an importance sampling fashion, to yield a cheaper estimate:

$$\log Z_\theta = \log \int e^{-E_\theta(x)} \, \mathrm{d}x = \log \int \frac{e^{-E_\theta(x)}}{q(x)} q(x) \, \mathrm{d}x \geq \mathbb{E}_{X_1, \ldots, X_M \sim q} \left[ \log \frac{1}{M} \sum_{m=1}^{M} \frac{e^{-E_\theta(X_m)}}{q(X_m)} \right], \tag{6}$$

where the last inequality is a consequence of Jensen's inequality (Jensen, 1905; 1906). In turn, this means that we will maximise the likelihood upper bound

$$\ell_{\text{IS}}(\theta) = \frac{1}{n} \sum_{i=1}^{n} -E_\theta(x_i) - \mathbb{E}_{X_1, \ldots, X_M \sim q} \left[ \log \frac{1}{M} \sum_{m=1}^{M} \frac{e^{-E_\theta(X_m)}}{q(X_m)} \right] \geq \ell(\theta), \tag{7}$$

in lieu of the likelihood. Depending on the choice of $q$ and on the number of importance samples $M$, this inequality is potentially very loose, meaning that one would train the model to maximise a biased approximation of the likelihood. Finding a good proposal $q$ that allows for fast sampling and correct estimation of its entropy is still a very active research area (Grathwohl et al., 2021; Kumar et al., 2019; Xie et al., 2018). Usually, this proposal is trained in parallel with the model $E_\theta$, which leads to a very unstable adversarial objective (Geng et al., 2021).

## 2.2 Can we make this logarithm disappear?

The looseness of the importance sampling approximation $\ell_{\mathrm{IS}}(\theta)$ is only due to Jensen's inequality: if the logarithm were replaced by a linear function, it would be possible to compute an unbiased estimate of the log-likelihood gradients. Our key idea is therefore to linearise the logarithm, using the following simple variational formulation. This will help us bypass the issues mentioned in Section 2.1.

**Lemma 2.1.** *For all $z > 0$,*

$$\log z = \min_{\lambda \in \mathbb{R}} \left( z e^{-\lambda} + \lambda - 1 \right). \tag{8}$$

The proof of this lemma is elementary and provided in Appendix A.1. This result is often used as an illustration of variational representations in variational inference tutorials (see, e.g., Jordan et al., 1999, Section 4.1; Ormerod & Wand, 2010, Section 3), but we are not aware of it being used in a context similar to ours. Applying Lemma 2.1 to Eq. (3) give us, for any $x \in \mathcal{X}$,

$$\begin{aligned}
\log p_\theta(x) &= -E_\theta(x) - \log Z_\theta = -E_\theta(x) - \min_{b \in \mathbb{R}} \left( e^{-b} Z_\theta + b - 1 \right) \\
&= -E_\theta(x) + \max_{b \in \mathbb{R}} \left( -e^{-b} Z_\theta - b + 1 \right) = \max_{b \in \mathbb{R}} \left( -E_\theta(x) - b - e^{-b} Z_\theta + 1 \right).
\end{aligned} \tag{9}$$

Using Eq. (9), we define a new objective named the **self-normalised log-likelihood (SNL)** $\ell_{\mathrm{SNL}}$ that is a function of the original parameter of the EBM $\theta$ and a single additional parameter $b \in \mathbb{R}$:

$$\ell_{\mathrm{SNL}}(b, \theta) = \frac{1}{n} \sum_{i=1}^{n} -E_\theta(x_i) - b - e^{-b} Z_\theta + 1. \tag{10}$$

When maximised w.r.t. $b$, we can recover the exact log-likelihood of a given model $p_\theta$ and maximising both $\theta$ and $b$ leads to the maximum log-likelihood estimate, as formalised below.

**Theorem 2.1.** *For any given $\theta$, when the SNL is maximised with respect to $b$, we have access to the exact log-likelihood of the model:*

$$\max_{b \in \mathbb{R}} \ \ell_{\mathrm{SNL}}(\theta, b) = \ell(\theta). \tag{11}$$

*Moreover, at the optimum, $b$ is the normalisation constant:*

$$\arg\max_{b \in \mathbb{R}} \ \ell_{\mathrm{SNL}}(\theta, b) = \log Z_\theta. \tag{12}$$

*Finally, there is a one-to-one correspondence between the local optima of the SNL and the log-likelihood.*

The proof is available in Appendix A.2 and is a simple application of the variational formulation of the logarithm. The important consequence of this result is that *maximising the SNL w.r.t. $\theta$ and $b$ will recover both the maximum log-likelihood estimate and its normalising constant.* This ability of our objective to learn both the model and its normaliser motivates the name **self-normalised log-likelihood**. We chose to call the extra parameter $b$ because, when $E_\theta$ is modelled as a neural network, $b$ can simply be understood as the bias of its last layer. In Appendix E.1, we propose another interpretation of SNL that derives directly from the Donsker-Varadhan variational representation of the KL distribution.

Another direct consequence of Eq. (9) is that, for any $\theta$ and $b$, SNL is a lower bound of the log-likelihood. Using the importance-sampling upper bound, this will lead to useful "sandwichings" of the log-likelihood:

$$\ell_{\mathrm{SNL}}(\theta, b) \leq \ell(\theta) \leq \ell_{\mathrm{IS}}(\theta). \tag{13}$$

### 2.3 Why maximising the SNL is easier

Why is the SNL more tractable than the standard log-likelihood? After all, the SNL also involves the intractable normalising constant. The key difference is that, since it depends linearly on it, it is now possible to obtain unbiased estimates of the SNL gradients.

Indeed, using a proposal $q$ gives us estimates of the gradient of $Z_\theta$ with importance sampling. Using

$$Z_\theta = \int \frac{e^{-E_\theta(x)}}{q(x)} q(x) dx = \mathbb{E}_{X \sim q} \left[ \frac{e^{-E_\theta(X)}}{q(X)} \right], \tag{14}$$

allows to get unbiased estimates of the SNL gradients w.r.t. $\theta$ and $b$. More precisely, for a batch of size $N_B$ and a number of samples $M$, we use the following estimate of the gradient w.r.t. $\theta$:

$$\begin{aligned}
\nabla_\theta \ell_{\text{SNL}}(\theta, b) &= -\frac{1}{n} \sum_{i=1}^n \nabla_\theta E_\theta(x_i) + e^{-b} \mathbb{E}_{X \sim q} \left[ \frac{\nabla_\theta E_\theta(X) e^{-E_\theta(X)}}{q(X)} \right] \\
&\approx -\frac{1}{n_B} \sum_{i=1}^{n_B} \nabla_\theta E_\theta(x_i) + e^{-b} \frac{1}{M} \sum_{m=1}^M \left[ \frac{\nabla_\theta E_\theta(x_m) e^{-E_\theta(x_m)}}{q(x_m)} \right].
\end{aligned} \tag{15}$$

Similarly, we can compute unbiased estimates of the gradients w.r.t. $b$:

$$\nabla_b \ell_{\text{SNL}}(\theta, b) = -1 + e^{-b} \mathbb{E}_{X \sim q} \left[ \frac{e^{-E_\theta(X)}}{q(X)} \right] \approx -1 + e^{-b} \frac{1}{M} \sum_{m=1}^M \left[ \frac{e^{-E_\theta(x_m)}}{q(x_m)} \right]. \tag{16}$$

The theory of stochastic optimisation (see, e.g., Bottou et al., 2018) then ensures that SGD-like algorithms, when applied to SNL, will converge to the maximum likelihood estimate and its normalising constant. In practice, we use popular algorithms like Adam (Kingma & Ba, 2015) to train $\theta$ and $b$ jointly. Some more specialised algorithms could also be used. For instance, Bietti & Mairal (2017) call optimisation problems similar to ours "infinite datasets with finite sum structure" (in our case, the infinite dataset is samples from the proposal, and the finite sum corresponds to the actual data), and propose an algorithm fit for this purpose. The full algorithm for training an energy-based model with SNL is available in Algorithm 1.

**Are the gradients of $\ell$ and $\ell_{\text{SNL}}$ related?** If we rewrite this gradient in the same fashion as Eq. (4), we can express the gradient of the SNL with the gradient of the log-likelihood:

$$\nabla_\theta \ell_{\text{SNL}}(\theta, b) = -\frac{1}{n} \sum_{i=1}^n \nabla_\theta E_\theta(x) - e^{-b + \log Z_\theta} \nabla \log Z_\theta = \nabla_\theta \ell_\theta + \nabla_\theta \log Z_\theta (1 - e^{-b + \log Z_\theta}). \tag{17}$$

When $b$ equals the normalisation constant $\log Z_\theta$, we obtain an unbiased estimator of the true log-likelihood gradient.

### 2.4 Practicalities when using SNL

For EBMs to be well-posed, it is required that the normalisation constant exists, i.e., that $\int e^{-E_\theta(x)} dx < \infty$. To that end, following Grathwohl et al. (2020) and similarly to exponential tilting (Siegmund, 1976), multiplying the un-normalised probability by a density $d$ ensures the existence of the normalisation constant. The distribution becomes $p_\theta(x) \propto e^{-E_\theta(x)} d(x)$.

We call $d$ the base density or the base distribution. In the case where the proposal $q$ is equal to the base distribution, the SNL estimates and the gradient estimates simplify:

$$\nabla_\theta Z_\theta \approx \frac{1}{M} \sum_{m=1}^M \nabla_\theta E_\theta(x_m) e^{-E_\theta(x_m)}. \tag{18}$$

Furthermore, we initialise $b$ by estimating $\log Z_\theta$ with importance sampling using the proposal $q$ at the beginning of the training procedure. This practice allows us to get gradient estimates of SNL somewhat close to the true gradient log-likelihood.

## 2.5 Related works

Objectives similar to SNL have been proposed in the past. In particular, in the context of local likelihood density estimation, Loader (1996) and Hjort & Jones (1996) handled intractable normalising constants in a similar fashion to ours. Arbel et al. (2020) leveraged a similar approach to estimate the normalisation constant to train hybrids of generative adversarial nets and EBMs. Neither of these works used importance sampling. Pihlaja et al. (2010) and Gutmann & Hirayama (2011) proposed families of generalisations of NCE which contain an objective similar to SNL as a special case. In these generalisations of NCE, the noise distribution plays a similar role to our proposal, but the obtained estimates in general differ from maximum likelihood. The novelty of SNL lies in the fact that it allows for performing exact maximum likelihood optimisation (regardless of the choice of proposal) for an EBM using stochastic optimisation together with importance sampling.

The SNL objective is related to the Donsker-Varadhan representation of the Kullback-Leibler divergence (Donsker & Varadhan, 1975), a variational formulation that has inspired several related approaches (Arbel et al., 2020; Belghazi et al., 2018; Glaser et al., 2021). SNL arises by *evaluating* the Donsker-Varadhan dual at a specific parametrisation $h(x) = -E_\theta(x) - b$, where $b$ learns the log-partition function (see Appendix E.1). Two closely related methods also build on this variational foundation but differ in key ways. The Generalized Energy Based Model (GEBM) (Arbel et al., 2020) uses the same parametrisation as SNL, but treats the base measure $\mathbb{B}$ as a *learnable* implicit model (e.g., a GAN generator), jointly optimising both the energy and the base via alternating updates. A key distinction is that in SNL, the base measure $d(x)$ defines the probabilistic model while the proposal $q(x)$ used for importance sampling can be chosen independently without affecting the optimum; in GEBM, no such separation exists, as samples for estimation are drawn directly from the learned base $\mathbb{B}$. KALE Flow (Glaser et al., 2021) takes a different approach, *optimising* over a restricted function class $\mathcal{H}$ with regularisation to define a surrogate divergence for gradient flows rather than maximum likelihood estimation. We discuss these connections and differences in detail in Appendices E.3 and E.4.

# 3 Some theoretical properties of SNL

## 3.1 Concavity of SNL for exponential families

It is a well-known fact that the log-likelihood of exponential families is concave because of the particular form of the gradient of the normaliser. We provide a proof in Appendix A.3 for completeness. The self-normalised log-likelihood preserves this property with the exponential family: the SNL is even *jointly* concave in both parameters.

**Theorem 3.1.** *If $(p_\theta)_\theta$ is a canonical exponential family, then $\ell_{\mathrm{SNL}}(\theta, b)$ is jointly concave.*

The proof is available in Appendix A.4 and follows directly from the convexity of the exponential. This means that the many theoretical results on stochastic optimisation for convex functions could be leveraged to prove convergence guarantees of SNL (see, e.g., Bottou et al., 2018).

## 3.2 An information-theoretic interpretation

Maximum likelihood has the following classical information-theoretic interpretation: when the number of samples goes to infinity, maximising the likelihood is equivalent to minimising the Kullback-Leibler divergence between $p_\theta$ and the true data distribution $p_{\mathrm{data}}$ (see, e.g., White, 1982). A similar rationale also exists for SNL and involves a generalisation of the Kullback-Leibler divergence to un-normalised finite measures. This generalisation exists also in the more general context of $f$-divergences, as detailed for instance by Amari & Nagaoka (2000, Section 3.6) or Stummer & Vajda (2010). It reduces to the usual definition when $f_1$ and $f_2$ are probability densities and shares many of the merits of the usual Kullback-Leibler divergence (see Appendix D for more details).

Standard maximum likelihood is asymptotically equivalent to minimising $\mathrm{KL}(p_{\mathrm{data}}||p_\theta)$. As we detail in Appendix D, this turns out to be equivalent to minimising the generalised divergence between $p_{\mathrm{data}}$ and all un-normalised models proportional to $e^{-E_\theta}$:

$$\mathrm{KL}(p_{\mathrm{data}}||p_\theta) = \min_{c>0} \mathrm{KL}(p_{\mathrm{data}}||ce^{-E_\theta}). \tag{19}$$

This new divergence is related to the SNL in the same way that the standard Kullback-Leibler divergence is related to the likelihood. Indeed, for any $c > 0$,

$$\mathrm{KL}(p_{\mathrm{data}}||ce^{-E_\theta}) = \int \log\left(\frac{p_{\mathrm{data}}(x)}{ce^{-E_\theta(x)}}\right) p_{\mathrm{data}}(x)(x)dx + cZ_\theta - 1 \tag{20}$$

$$= -\int e^{-E_\theta(x)} p_{\mathrm{data}}(x)dx - \log c + cZ_\theta - 1 + \underbrace{\int \log(p_{\mathrm{data}}(x))p_{\mathrm{data}}(x)dx}_{\text{does not depend on } \theta \text{ nor } c}. \tag{21}$$

The first integral, that depends on $\theta$, is intractable, but may be estimated if we have access to an i.i.d. dataset $x_1, \ldots, x_n$, leading to the estimate

$$\mathrm{KL}(p_{\mathrm{data}}||ce^{-E_\theta}) \approx -\frac{1}{n}\sum_{i=1}^{n} e^{-E_\theta(x_i)} - \log c + cZ_\theta - 1 + \int \log(p_{\mathrm{data}}(x))p_{\mathrm{data}}(x)\,\mathrm{d}x \tag{22}$$

$$= -\ell_{\mathrm{SNL}}(\theta, \log c) + \int \log(p_{\mathrm{data}}(x))p_{\mathrm{data}}(x)\,\mathrm{d}x, \tag{23}$$

which means that minimising the SNL will asymptotically resemble minimising the generalised Kullback-Leibler divergence. In the context of local likelihood density estimation, Hjort & Jones (1996) also derived similar connections with the generalised Kullback-Leibler divergence. More recently, Bach (2022) applied the same variational representation of the logarithm to the generalised Kullbakc-Leibler, in a context very different from ours.

## 4 Extending SNL beyond basic density estimation

### 4.1 Truncated densities

Truncated densities are probability density functions defined on truncated domains. They retain the same parametric form as their non-truncated counterparts, differing only by a normalising constant. However, because this normalising constant is often difficult or impossible to compute analytically, applying Maximum Likelihood Estimation to truncated density models becomes challenging. Even for a simple Gaussian distribution with more than two dimensions, estimating such parameters is not straightforward. We restrict the study to the truncated density of the following shape:

$$p_\theta^t(x) = \frac{t(x)p_\theta(x)}{Z_\theta^t} \tag{24}$$

where $p_\theta(x)$ is a standard density, $t : \mathcal{X} \to \{1, 0\}$ is the **known** truncation function and $Z_\theta^t$ is the unknown normalisation constant. The corresponding energy function for an EBM is then

$$E_\theta^t(x) = \begin{cases} -\log p_\theta(x) & \text{if } t(x) = 1, \\ -\infty & \text{else.} \end{cases} \tag{25}$$

### 4.2 Self-normalisation in the regression setting

We consider the supervised regression problem where we are given a dataset of pairs of inputs and targets $(x, y) \in \mathcal{X} \times \mathcal{Y}$ where the target space $\mathcal{Y}$ is continuous. We want to estimate the conditional distribution $p_{\mathrm{data}}(y|x)$ using an EBM:

$$p_\theta(y|x) = \frac{e^{-E_\theta(x,y)}}{Z_{\theta,x}}, \tag{26}$$

where $Z_{\theta,x} = \int e^{-E_\theta(x,y)}\,\mathrm{d}y$. The main difference with the previous density estimation setup is that the normalisation constant $Z_{\theta,x}$ also depends on the input value $x$.

Because the normaliser now also depends on $x$, we introduce a new family of functions $b_\phi$ whose role is to estimate the normalisation constant $Z_{\theta,x}$. Similarly to the density estimation case, we define the self-normalised log-likelihood as

$$\ell_{\mathrm{SNL}}(\theta, \phi) = \frac{1}{n}\sum_{i=1}^{n}\left(-E_\theta(x_i, y_i) - b_\phi(x_i) - Z_{\theta,x_i}e^{-b_\phi(x_i)} + 1\right). \tag{27}$$

Provided the family $b_\phi$ is expressive enough, this SNL for regression enjoys the same properties as its unsupervised counterpart. We can retrieve the maximum likelihood estimate when maximising the SNL in both $\theta$ and $\phi$. Moreover, at the optimum, for any $x \in \mathcal{X}$, $b_\phi(x)$ is the normalisation constant $\log Z_{\theta,x}$. The SNL for regression is also a lower bound of the true conditional log-likelihood. Following the reasoning of Section 2.3, we propose to train an EBM model for regression using the SNL. To that end, we consider a proposal $q_\psi$ that depends on both $x$ and $y$ and is parameterised by $\psi$. For instance, Gustafsson et al. (2020) use a mixture density network (MDN, Bishop, 1994) proposal. In the work of Gustafsson et al. (2020), the EBM is trained jointly with the MDN. The MDN maximisation objective is an average combination between the negative Kullback-Leibler divergence between the $p_\theta$ and $q_\psi$.

In concurrent work, Sander et al. (2025) introduced the SNL loss of Eq. (27) for regression in the context of multi-label classification and label ranking. They extend Theorem 2.1 to the same neural network parametrisation of the normalisation constant $b_\phi$ and study the connection to broader Fenchel-Young losses and the generalisation capacity of the objective.

### 4.3 Self-normalised evidence lower bound

The SNL approach allows training a variational auto-encoder (VAE, Kingma & Welling, 2014) with an energy-based prior using approximate inference. Both Pang et al. (2020) and **?** trained an EBM as a prior in the latent space for a noisy sampler, but required MCMC to sample from the posterior and the prior during training. We introduce the self-normalised evidence lower bound (SNELBO), a surrogate ELBO objective that leverages the self-normalised log-likelihood to allow for straightforward training.

Formally, we consider a VAE with a prior $p_\theta(z)$ defined by an EBM composed of an energy function $E_\theta$ parameterised by a neural network and an associated base distribution $d(z)$, i.e., $p_\theta(z) = \frac{e^{-E_\theta}d(z)}{Z_\theta}$ where $Z_\theta = \int e^{-E_\theta(z)}d(z)dz$. The generative model is the same as in VAE, and an output density $p_\phi(x|z)$ is parameterised by a neural network $g_\phi(z)$. Since the likelihood is intractable, we posit a conditional variational distribution $q_\gamma(z|x)$ to approximate the posterior of the model, similarly to the original VAE. Using Lemma 2.1, we can obtain the SNELBO:

$$\mathcal{L}_{\mathrm{SNL}}(\theta, \phi, \gamma, b) = \mathbb{E}_{q_\gamma(z|x)}\left[\log p_\phi(x|z)\right] + \mathbb{E}_{q_\gamma(z|x)}\left[\log \frac{d(z)}{q_\gamma(z|x)}\right]$$
$$+ \mathbb{E}_{q_\gamma(z|x)}\left[-E_\theta(z) - b\right] - \mathbb{E}_{d(z)}\left[e^{-E_\theta(z)-b}\right] + 1. \quad (28)$$

We note that the SNELBO is a lower bound on the log-likelihood, $\ell(\theta, \phi)$, and a lower bound on the regular ELBO, $\mathcal{L}$, that is tight for optimal $b$, i.e., $\ell(\theta, \phi) \geq \mathcal{L}(\theta, \phi, \gamma) \geq \mathcal{L}_{\mathrm{SNL}}(\theta, \phi, \gamma, b)$ and $\mathcal{L}(\theta, \phi, \gamma) = \max_{b \in \mathbb{R}} \mathcal{L}_{\mathrm{SNL}}(\theta, \phi, \gamma, b)$. See Appendix G for derivation details. This surrogate objective can be interpreted as the combination of the ELBO from a VAE whose prior is the base distribution $d(z)$ with a regularisation term from the EBM. As such, the EBM can be added easily on top of any VAE model.

## 5 Experiments

In this section, we employ SNL for different applications. While SNL does not achieve state-of-the-art performance in every case, it provides an easy-to-use, out-of-the-box solution for problems where the normalising constant is unavailable. Although extending SNL to high-dimensional, arbitrary density estimation remains challenging, it works seamlessly for directional and truncated distributions, which are otherwise nontrivial cases. Moreover, we obtain state-of-the-art results with SNL applied to energy-based models (EBMs) for image regression.

### 5.1 Density estimation

### 5.1.1 Density estimation for directional distributions

As presented Theorem 3.1, the SNL objective is concave for exponential families. This makes the objective very attractive for exponential families with unknown normalisation. This includes many directional distributions, such as the multivariate von Mises (Mardia et al., 2008) for which no tractable formulas exist. The multivariate von Mises distribution has a density of the form

$$\mathcal{M}v\mathcal{M}(x) = \frac{1}{Z_{\theta,\mathbf{\Lambda},\boldsymbol{\kappa}}} \exp\left(\boldsymbol{\kappa}^T c(x, \theta) + s(x, \theta)^T \mathbf{\Lambda} s(x, \theta)\right), \quad (29)$$

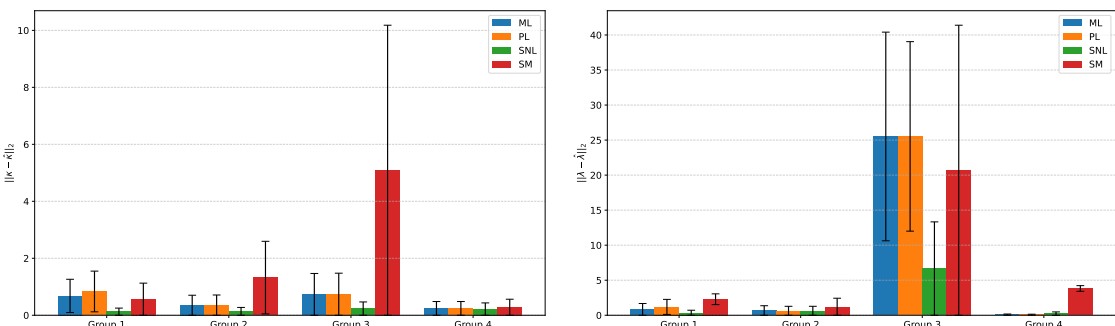

Figure 2: Error on the estimated parameters obtained using different likelihood methods (approximate maximum likelihood, ML), pseudo likelihood (PL), self-normalising likelihood (SNL) and score matching (SM, Mardia et al., 2008). The score and uncertainty for ML, PL and MM are directly reported from Mardia et al. (2008) while we report average scores of SNL and SM over five different generated datasets and runs. The exact value of the estimated parameters is given in Table 17 of the appendix.

with $Z_{\theta,\boldsymbol{\Lambda},\boldsymbol{\kappa}}$ an unknown normalisation constant, $\theta$ the localisation parameter, $\boldsymbol{\Lambda}$ the correlation parameter and $\boldsymbol{\kappa}$ the concentration parameters. These parameters satisfy $-\pi < \theta_i \leq \pi$, $-\pi < \mu_i \leq \pi$ and $\boldsymbol{\kappa}_i \geq 0$. The matrix $\boldsymbol{\Lambda}$ real valied and symmetric with $(\boldsymbol{\Lambda})_{ij} = \lambda_{ij} = \lambda_{ji}$ for $i \neq j$ and diagonal elements $\lambda_{ii} = 0$. We defined $c(x,\theta)^T = \left(\cos\left(x - \theta_1\right), \cos\left(x - \theta_2\right), \ldots, \cos\left(x - \theta_p\right)\right)$ and $s(x,\theta)^T = \left(\sin\left(x - \theta_1\right), \sin\left(x - \theta_2\right), \ldots, \sin\left(x - \theta_p\right)\right)$.

We evaluate the performance of our SNL estimator using datasets sampled with four different sets of parameters of the multivariate von Mises distribution, following the setup of Mardia et al. (2008). Following their experimental set-up, we fix the localisation parameters $\theta$ to their true value and only estimate the concentration and correlation parameters. Each set of parameters was designed to explore the possible combinations with high/low concentration $\kappa$ and high/low correlation $\lambda_{ij}$. In Figure 2, we compare our estimator to score matching (SM) by Mardia et al. (2016), pseudo-likelihood and approximate maximum likelihood with numerical integration by Mardia et al. (2008). SNL outperforms the other estimators in most cases, except in the low correlation case where the variance is high.

### 5.1.2 Density estimation for truncated distributions

We evaluate the performance of SNL on a truncated distribution using a simple test without model misspecification from Melchior & Goulding (2018). We draw samples from a Gaussian Mixture Model with $K = 3$ clusters truncated by a box and a circle with known boundaries. As opposed to the original implementation by Melchior & Goulding (2018), we do not add noise to the sample and simply discard samples outside these boundaries to create the truncated distribution. The details of our parameterisation are provided in Appendix I.2.

We evaluate the log-likelihood of the resulting model (without truncation) on the non-truncated test dataset in Fig. 3. We further evaluate the quality of the estimation by comparing the parameters of the estimated mixture to the true parameters in Fig. 8. Although the performance in likelihood does not match the specialised GMMis implementation of Melchior & Goulding (2018), the simplicity of our approach suggests that SNL constitutes a promising and flexible alternative for truncated density estimation.

### 5.1.3 Density estimation for tabular data

In this section, we evaluate the capacity of SNL to train an EBM for density estimation. We consider both an artificial dataset and a real dataset from UCI. In both cases, we compare our model with an EBM trained in the same condition but with noise contrastive estimation (NCE, Gutmann & Hyvärinen, 2010).

We evaluate the performances of EBMs for density estimation, trained with SNL on four different, two-dimensional, generated datasets. We compare our model with an EBM trained in the same condition but with noise contrastive estimation. Both setup also leverages a base distribution that equals the proposal distribution. Qualitatively in Fig. 6, we observe that the two models perform on par, except for the four

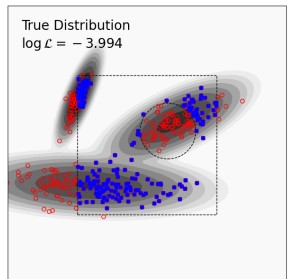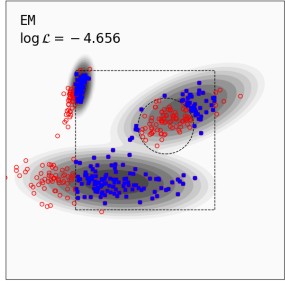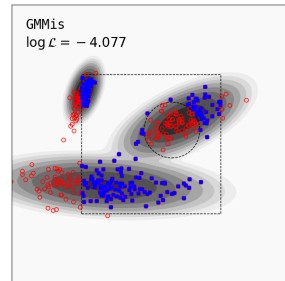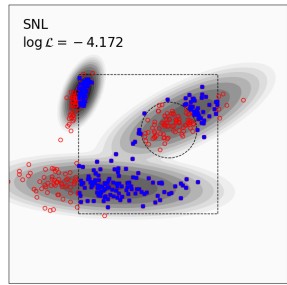

Figure 3: Different methods are used to estimate the parameters of a truncated mixture of 3 Gaussians. Each method is fed with truncated data (the blue points). The log-likelihood $\mathcal{L}$ is evaluated on a complete test set (i.e. without truncation) in order to verify the quality of the estimation. From left to right, we show the true density of the GMM, the model obtained with a standard Expectation maximisation algorithm, the model obtained with an EM using imputation (GMMis Melchior & Goulding (2018)) and our model trained using gradient descent and SNL.

| | | Funnel | | Pinwheel | | Checkerboard | | Four Circles | |
|---|---|---|---|---|---|---|---|---|---|
| Objective | Base Dist | $\ell_{\mathrm{IS}}$ | $\ell_{\mathrm{SNL}}$ | $\ell_{\mathrm{IS}}$ | $\ell_{\mathrm{SNL}}$ | $\ell_{\mathrm{IS}}$ | $\ell_{\mathrm{SNL}}$ | $\ell_{\mathrm{IS}}$ | $\ell_{\mathrm{SNL}}$ |
| NCE | $\mathcal{N}(\mathbf{0},\mathbf{1})$ | $-2.040$ $_{(\pm 0.251)}$ | $-2.044$ $_{(\pm 0.254)}$ | $-\mathbf{1.004}$ $_{(\pm 0.072)}$ | $-\mathbf{1.020}$ $_{(\pm 0.084)}$ | $-1.947$ $_{(\pm 0.033)}$ | $-1.964$ $_{(\pm 0.032)}$ | $-2.117$ $_{(\pm 0.005)}$ | $-2.120$ $_{(\pm 0.006)}$ |
| SNL | $\mathcal{N}(\mathbf{0},\mathbf{1})$ | $-\mathbf{1.811}$ $_{(\pm 0.175)}$ | $-\mathbf{1.831}$ $_{(\pm 0.181)}$ | $-1.031$ $_{(\pm 0.066)}$ | $-1.035$ $_{(\pm 0.065)}$ | $-\mathbf{1.902}$ $_{(\pm 0.012)}$ | $-\mathbf{1.905}$ $_{(\pm 0.012)}$ | $-\mathbf{1.914}$ $_{(\pm 0.022)}$ | $-\mathbf{1.918}$ $_{(\pm 0.022)}$ |
| NCE | None | $-1.894$ $_{(\pm 0.096)}$ | $-1.896$ $_{(\pm 0.097)}$ | $-1.063$ $_{(\pm 0.019)}$ | $-1.069$ $_{(\pm 0.024)}$ | $-1.997$ $_{(\pm 0.022)}$ | $-2.025$ $_{(\pm 0.056)}$ | $-2.231$ $_{(\pm 0.038)}$ | $-2.232$ $_{(\pm 0.039)}$ |
| SNL | None | $-2.006$ $_{(\pm 0.378)}$ | $-2.066$ $_{(\pm 0.468)}$ | $-1.072$ $_{(\pm 0.040)}$ | $-1.086$ $_{(\pm 0.030)}$ | $-1.966$ $_{(\pm 0.030)}$ | $-1.969$ $_{(\pm 0.028)}$ | $-1.971$ $_{(\pm 0.047)}$ | $-1.973$ $_{(\pm 0.048)}$ |

Table 1: Evaluation of the performance of EBMs trained with NCE or SNL objective with or without a base distribution. We generate each dataset five times and run each set of parameters once on each. We report the mean and standard deviation of the estimated log-likelihood and the self-normalised likelihood $\ell_{SNL}$. Highest is best.

circles dataset, where SNL dominates. We explore the impact of the base distribution in this setting. We show our results in Table 1. According to those results, SNL-trained EBMs with a base distribution perform better than all the NCE settings across all datasets except Pinwheel. Using a base distribution always improves the performance with SNL, while it varies with NCE.

We also evaluate SNL-trained EBMs on UCI datasets in order to assess the impact of increasing dimensionality. We use a simple Gaussian proposal with full covariance. The results, reported in Table 2, show that EBM-SNL is competitive on lower-dimensional tabular datasets, while its performance degrades compared to strong normalizing-flow baselines on Hepmass ($d = 21$). Despite this drop in performance at higher dimensions, the method remains appealing in low-dimensional settings due to its simplicity, relying only on a fully connected neural network for the EBM and a simple Gaussian proposal.

## 5.2 EBMs for regression

Following Gustafsson et al. (2022), we study and compare our training method on two one-dimensional regression tasks (seeFig. 5 in Appendix H) and four image regression datasets. We parameterise our model with the same architecture as Gustafsson et al. (2022) where the output of a feature extractor, $h_x$, feeds both the proposal $q_\psi(.|h_x)$ and a head neural network for the EBM (see Figure 1 in Gustafsson et al. (2022) for more details). When used as a proposal, the weights of the feature extractor are frozen.

| | EBM - SNL | Gaussian | MADE | MADE MoG | Real NVP (5) | Real NVP (10) | MAF (5) | MAF (10) | MAF MoG (5) |
|---|---|---|---|---|---|---|---|---|---|
| Power ($d = 6$) | $[0.28, 0.41]$ | $-7.74$ | $-3.08$ | $0.40$ | $-0.02$ | $0.17$ | $0.14$ | $0.24$ | $0.30$ |
| Gas ($d = 8$) | $[5.73, 7.74]$ | $-3.58$ | $3.56$ | $8.47$ | $4.78$ | $8.33$ | $9.07$ | $10.08$ | $9.59$ |
| Hepmass ($d = 21$) | $[-19.22, -19.20]$ | $-27.93$ | $-20.98$ | $-15.15$ | $-19.62$ | $-18.71$ | $-17.70$ | $-17.73$ | $-17.39$ |

Table 2: For EBM-SNL the upper bound corresponds to $\ell_{IS}$ and the lower bound to $\ell_{SNL}$. Both are computed using 20000 samples from the test set.

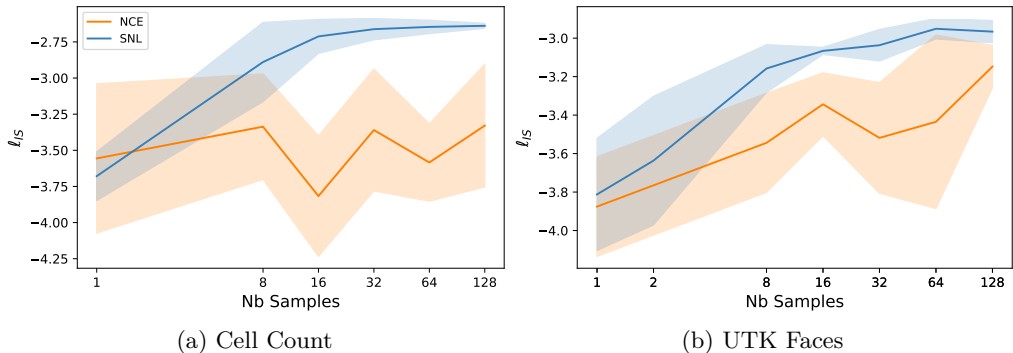

(a) Cell Count  (b) UTK Faces

Figure 4: This graphs depict how the number of samples $M$ in the proposal affects the performance of the EBM for regression. Each model is trained on the Cell Count and UTKFaces dataset with a Gaussian proposal.

In our experiments, we consider three different proposals:

- A mixture density network proposal whose parameters are given by a small fully connected neural network.

- A fixed multivariate Gaussian $\mathcal{N}(\mu, \Sigma)$ whose parameters are estimated before training with the training dataset and fixed during training.

- A fixed uniform distribution $\mathcal{U}$ that is defined by leveraging the knowledge from the dataset and fixed during training.

All models are evaluated using an estimate of the log-likelihood with $M = 20{,}000$ samples $y_i^{(m)}$ from a multivariate Gaussian whose parameters are estimated before training:

$$\ell_{IS} = \frac{1}{N} \sum_{i=1}^{N} \left( -E_\theta(x_i, y_i) - b_\phi(x_i) - \log \frac{1}{M} \sum_{m=1}^{M} e^{-E_\theta(x_i, y_i^{(m)}) - b_\phi(x_i)} \right). \tag{30}$$

Since NCE normalises the EBM at the optimum (Mnih & Teh, 2012), we also provide the $\ell_{SNL}$ (i.e. a lower bound estimate of the log-likelihood) for each set of parameters using the same proposal as $\ell_{IS}$ with 20,000 samples. If $\ell_{SNL}$ is close to $\ell_{IS}$, this means that the lower-bound is tight and $b_\phi$ approximates correctly $\log Z_\theta$ (or if no $b_\phi$ is used the network is self-normalised and $Z_\theta = 1$).

### 5.2.1  1D regression datasets

We consider here the two artificial datasets for 1D regression with multimodal distributions $p(y|x)$ (see Fig. 5). We provide a description of the neural network architecture in Appendix I.4.1. On both datasets, the SNL always outperformed its NCE counterparts with respect to the estimated upper bound $\ell_{IS}$, as seen in Table 3. Moreover, the $\ell_{SNL}$ of the NCE is loose compared to the $\ell_{SNL}$. This is due to a poor estimation of the normalisation constant with NCE. We provide additional results in Table 20. Using a base distribution to ensure the existence of the normalisation constant $Z_\theta$ either improves or gives similar results with the SNL objective but systematically damages the results when minimising the NCE loss. As mentioned by Mnih & Teh (2012), with both objectives, explicitly modelling $b_\phi$ does not provide a better estimation of the network. The normalisation is implicitly learned with $E_\theta$.

### 5.2.2  Image regression datasets

We train an NCE-EBM setup and an SNL-EBM setup on an image regression task. We train on four different datasets, steering angle, cell count, UTKFaces and BIWI and follow the same setup as Gustafsson et al. (2022). Similarly to the 1D regression datasets, SNL-trained EBM always outperforms its NCE counterparts (Table 4). When using NCE, the normalisation constant is off resulting in a loose lower bound of the likelihood whereas SNL usually provides a better approximation. In Fig. 4, we observe that our method improves with

| Objective | Proposal $q$ | Regression Dataset 1 | | Regression Dataset 2 | |
|---|---|---|---|---|---|
| | | $\ell_{\text{IS}}$ | $\ell_{\text{SNL}}$ | $\ell_{\text{IS}}$ | $\ell_{\text{SNL}}$ |
| NCE | $\mathcal{N}(\mu, \Sigma)$ | $-0.030$ $(\pm 0.278)$ | $-0.718$ $(\pm 0.256)$ | $-2.592$ $(\pm 0.214)$ | $-3.559$ $(\pm 1.881)$ |
| NCE | MDN K2 | $-0.611$ $(\pm 0.154)$ | $-1.492$ $(\pm 0.993)$ | $-2.451$ $(\pm 0.088)$ | $-2.634$ $(\pm 0.084)$ |
| SNL | $\mathcal{N}(\mu, \Sigma)$ | $0.164$ $(\pm 0.088)$ | $0.033$ $(\pm 0.077)$ | $\mathbf{-1.813}$ $(\pm 0.109)$ | $\mathbf{-1.836}$ $(\pm 0.109)$ |
| SNL | MDN K2 | $\mathbf{0.255}$ $(\pm 0.017)$ | $\mathbf{0.251}$ $(\pm 0.016)$ | $-2.099$ $(\pm 0.250)$ | $-2.170$ $(\pm 0.353)$ |

Table 3: Evaluation of regression EBMs on the 1D toy regression problems with two different objectives and two different proposals. Each model is trained for five runs and we report the mean and standard deviation of the estimated log-likelihood $\ell_{\text{IS}}$ and the self normalized log-likelihood $\ell_{\text{SNL}}$. Using the SNL as objective clearly outperforms the NCE.

| Objective | Proposal | Steering Angle | | Cell Count | | UTKFaces | | BIWI | |
|---|---|---|---|---|---|---|---|---|---|
| | | $\ell_{\text{IS}}$ | $\ell_{\text{SNL}}$ | $\ell_{\text{IS}}$ | $\ell_{\text{SNL}}$ | $\ell_{\text{IS}}$ | $\ell_{\text{SNL}}$ | $\ell_{\text{IS}}$ | $\ell_{\text{SNL}}$ |
| NCE | $\mathcal{N}(\mu, \Sigma)$ | $-3.649$ $(\pm 1.224)$ | UNNORMALIZED | $-3.367$ $(\pm 0.399)$ | $-9.675$ $(\pm 0.605)$ | $-3.147$ $(\pm 0.1100)$ | $-8.223$ $(\pm 3.795)$ | $-11.02$ $(\pm 0.576)$ | UNNORMALIZED |
| NCE | MDN-8 | $-4.001$ $(\pm 0.667)$ | UNNORMALIZED | $-3.864$ $(\pm 0.048)$ | UNNORMALIZED | $-4.123$ $(\pm 0.21)$ | $-5.170$ $(\pm 0.955)$ | $-11.998$ $(\pm 0.339)$ | UNNORMALIZED |
| SNL | $\mathcal{N}(\mu, \Sigma)$ | $-2.665$ $(\pm 1.37)$ | $-3.973$ $(\pm 3.15)$ | $-2.701$ $(\pm 0.041)$ | $-2.725$ $(\pm 0.046)$ | $-2.966$ $(\pm 0.057)$ | $-2.991$ $(\pm 0.069)$ | $-10.86$ $(\pm 1.017)$ | $-11.05$ $(\pm 1.141)$ |
| SNL | Uniform | $\mathbf{-1.402}$ $(\pm 0.068)$ | $\mathbf{-1.423}$ $(\pm 0.074)$ | $\mathbf{-2.604}$ $(\pm 0.001)$ | $\mathbf{-2.620}$ $(\pm 0.007)$ | $-2.927$ $(\pm 0.032)$ | $\mathbf{-2.965}$ $(\pm 0.019)$ | $-10.44$ $(\pm 0.138)$ | $-10.51$ $(\pm 1.222)$ |
| SNL | MDN-8 | $-1.673$ $(\pm 0.042)$ | $-1.692$ $(\pm 0.046)$ | $-2.801$ $(\pm 0.071)$ | $-2.811$ $(\pm 0.071)$ | $\mathbf{-2.921}$ $(\pm 0.055)$ | $-2.943$ $(\pm 0.062)$ | $\mathbf{-10.01}$ $(\pm 0.092)$ | $\mathbf{-10.04}$ $(\pm 0.091)$ |

Table 4: Evaluation of EBMs for regression on image regression datasets with two different objectives and different proposals. Each model is trained for five runs, and we report the mean and standard deviation of the estimated log-likelihood ($\ell_{IS}$) and estimated self-normalised log-likelihood ($\ell_{SNL}$). When the proposal is MDN, the proposal is learned jointly with the model following Gustafsson et al. (2022).

the number of samples but stagnates after $M = 64$ samples. On the other hand, NCE seems to improve with the sample size but in a less compelling fashion. We provide additional results in Table 21.

### 5.3 VAE with latent prior EBM

We train a VAE with EBM prior on binary MNIST using SNELBO, as outlined in Section 4.3. We parameterise the output distribution with a Bernoulli distribution with parameters from a neural network $g_\phi$, i.e. $p_\phi(x|z) = \mathcal{B}(x|g_\phi(z))$ and the approximate posterior with a Gaussian whose parameters are given by a neural network $q_\gamma(z|x)$. We either train from scratch the VAE with EBM prior (VAE-EBM) or we only train the prior of a pre-trained VAE with a standard Gaussian prior (VAE-EBM Post-hoc). We compare to a standard VAE with a Gaussian prior (VAE) and a VAE with a Mixture of Gaussian Prior (VAE-MoG) and 10 mixtures. All VAEs are trained with a latent space of size 16. In Table 5, we show that training VAE-EBM with latent EBM provides better SNELBO.

| | |
|---|---|
| VAE | -89.10 |
| VAE-MoG | -88.73 |
| VAE-EBM Post-Hoc | -88.11 |
| VAE-EBM | **-87.09** |

Table 5: ELBO/SNELBO for VAEs with different priors.

We report the FID scores of generated samples for a standard VAE, a VAE with a learned EBM prior, and a latent short-term MCMC approach Pang et al. (2020) in Table 19. While the learned EBM prior yields only a marginal improvement over the vanilla VAE, the latent short-term MCMC method produces substantially higher-quality samples. This improvement, however, comes at the cost of losing direct access to ELBO or SNELBO estimates.

## 6 Conclusion

We proposed a new objective to train energy-based models (EBMs) called self-normalising log-likelihood (SNL). By maximising SNL with respect to the parameters of the EBM and an additional single parameter $b$, we can recover both the maximum likelihood estimate and the normalising constant at optimality. We conducted an extensive experimental study on parameter estimation for directional and truncated distributions, low-dimensional datasets for density estimation, complex regression problems and training VAEs with EBM prior. These experiments illustrate that SNL can easily handle intractable normalising constants in many different situations. However, in high-dimensional settings, the variance of the importance sampling estimates used to approximate the partition function and its gradients grows rapidly, which limits the practical scalability of

the method. Reducing this variance, for instance through more expressive proposals or variance reduction techniques, will be the subject of future work.

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

## A  Proofs

### A.1  Lemma 2.1: Variational linearising the logarithm

Inspired by Jordan et al. (1999) and Ormerod & Wand (2010), we show the following lemma:

**Lemma 2.1.** *For all $z > 0$,*

$$\log z = \min_{\lambda \in \mathbb{R}} \left( z e^{-\lambda} + \lambda - 1 \right). \tag{8}$$

*Proof.* Let $z > 0$ and $\lambda \in \mathbb{R}$, we define the function:

$$h(\lambda) = z e^{-\lambda} + \lambda - 1. \tag{31}$$

By differentiating this function with respect to $\lambda$, we get:

$$h'(\lambda) = -z e^{-\lambda} + 1. \tag{32}$$

The differentiated function $h'$ is negative for $\lambda < \log z$ and positive for $\lambda > \log z$. Thus the minimum of $h$ is reached at $\lambda = \log(z)$ and $h(\log(z)) = \log(z)$, hence the proof. $\square$

### A.2  Theorem 2.1: Equivalence between SNL and the log-likelihood

We begin by reminding notations: we consider an energy-based model, which specifies a probability density over a space $\mathcal{X}$ through a parameterised energy function $E_\theta : \mathcal{X} \to \mathbb{R}$. The associated density is:

$$p_\theta(x) = \frac{e^{-E_\theta(x)}}{Z_\theta}, \tag{33}$$

where $Z_\theta = \int e^{-E_\theta}(x) \mathrm{d}x$ is the partition function. Given $n$ data points $x_1, ..., x_n \in \mathcal{X}$, we define the log-likelihood function:

$$\ell(\theta) = \frac{1}{n} \sum_{i=1}^{n} -E_\theta(x_i) - \log Z_\theta. \tag{34}$$

We define as well the self-normalised log-likelihood (SNL) as:

$$\ell_{\mathrm{SNL}}(\theta, b) = \frac{1}{n} \sum_{i=1}^{n} -E_\theta(x_i) - b - e^{-b} Z_\theta + 1. \tag{35}$$

We now recall Theorem 2.1:

**Theorem 2.1.** *For any given $\theta$, when the SNL is maximised with respect to $b$, we have access to the exact log-likelihood of the model:*

$$\max_{b \in \mathbb{R}} \ell_{\text{SNL}}(\theta, b) = \ell(\theta). \tag{11}$$

*Moreover, at the optimum, $b$ is the normalisation constant:*

$$\arg\max_{b \in \mathbb{R}} \ell_{\text{SNL}}(\theta, b) = \log Z_\theta. \tag{12}$$

*Finally, there is a one-to-one correspondence between the local optima of the SNL and the log-likelihood.*

*Proof.* Using Lemma 2.1, we show that for any $\theta$, $\max_{b \in \mathbb{R}} \ell_{\text{SN}}(\theta, b) = \ell(\theta)$.

$$\ell(\theta) = \frac{1}{n} \sum_{i=1}^{n} \log p_\theta(x_i) \tag{36}$$

$$= \frac{1}{n} \sum_{i=1}^{n} \log(e^{-E_\theta(x_i)}) - \log Z_\theta \tag{37}$$

$$= \frac{1}{n} \sum_{i=1}^{n} \log(e^{-E_\theta(x_i)}) - \min_{b \in \mathbb{R}}(e^{-b} Z_\theta + b - 1) \tag{38}$$

$$= \frac{1}{n} \sum_{i=1}^{n} \log(e^{-E_\theta(x_i)}) + \max_{b \in \mathbb{R}}(-e^{-b} Z_\theta - b + 1) \tag{39}$$

$$= \max_{b} \frac{1}{n} \sum_{i=1}^{n} \log(e^{-E_\theta(x_i)}) - e^{-b} Z_\theta - b + 1 \tag{40}$$

$$= \max_{b \in \mathbb{R}} \ell_{\text{SNL}}(\theta, b). \tag{41}$$

We show that $\ell(\theta)$ and $\ell_{SNL}(\theta, b)$ share the same local maxima. Let $\theta^*$ a local optimum of $\ell$, we will construct a local optimum of $\ell_{SNL}$. Let $b^* = \log Z_{\theta^*}$, then

$$\nabla_b \ell_{SNL}(\theta^*, b^*)(x) = -1 + Z_{\theta^*} e^{-b} = 0. \tag{42}$$

Moreover,

$$\nabla_\theta \ell_{SNL}(\theta^*, b^*) = -\frac{1}{n} \sum_{i=1}^{n} \nabla_\theta E_\theta(x_i)|_{\theta^*} - e^{-b^*} \nabla_\theta Z_\theta|_{\theta^*} \tag{43}$$

$$= -\frac{1}{n} \sum_{i=1}^{n} \nabla_\theta E_\theta(x_i)|_{\theta^*} - e^{-b^*} Z_\theta \nabla_\theta \log Z_\theta|_{\theta^*} \tag{44}$$

$$= -\frac{1}{n} \sum_{i=1}^{n} \nabla_\theta E_\theta(x_i)|_{\theta^*} - \frac{1}{Z_{\theta^*}} Z_{\theta^*} \nabla_\theta \log Z_\theta|_{\theta^*} \tag{45}$$

$$= -\frac{1}{n} \sum_{i=1}^{n} \nabla_\theta E_\theta(x_i)|_{\theta^*} - \nabla_\theta \log Z_\theta|_{\theta^*} \tag{46}$$

$$= \nabla_\theta \ell(\theta^*) \tag{47}$$

$$= 0. \tag{48}$$

Thus, for any local optimum of $\ell(\theta^*)$, the pair $(\theta^*, \log Z_{\theta^*})$ is a local optimum $\ell_{SNL}$. Conversely, with the same reasoning, for any $(\tilde{\theta}, \tilde{b})$ local optimum of $\ell_{SNL}$, $\tilde{\theta}$ is a local maximum of $\ell$. $\qquad\square$

### A.3 Theorem A.1: Concavity of the log-likelihood in exponential families

For completeness, we prove the classical result about the convexity of exponential families. For more details, see e.g., Wainwright & Jordan (2008, Chapter 3).

**Theorem A.1.** *If $(p_\theta)_\theta$ is a canonical exponential family, then $\ell(\theta)$ is concave. In particular, the gradient and the Hessian of $\log Z_\theta$ are respectively the mean and the covariance matrix of the sufficient statistics.*

*Proof.* Let us consider an exponential family whose densities with respect to a base measure are parameterised as $p_\theta(x) = e^{\theta^T s(x) - \log Z_\theta}$, where $s(x)$ is the sufficient statistics and $\theta$ the natural parameters. To simplify formulas, we will assume that we observe a single data point $x \in \mathcal{X}$. Observing several i.i.d. data points will preserve concavity because a sum of concave functions remains concave, so there is no loss of generality. The log-likelihood of such a model is given by:

$$\ell(\theta) = \theta^T s(x) - \log Z_\theta = \theta^T s(x) - \log \int e^{\theta^T s(x)} \mathrm{d}x. \tag{49}$$

We will prove that this objective is concave by showing that the Hessian is negative semi-definite. Let's calculate the gradient and Hessian of $\log Z_\theta$. For integrals akin to the normalising constant, switching differentiation and integration is allowed (see, e.g., Lehmann & Romano, 2022, Theorem 2.7.1), and we get

$$\nabla_\theta \log Z_\theta = \nabla_\theta \log \int e^{\theta^T s(x)} dx \tag{50}$$

$$= \frac{\int s(x) e^{\theta^T s(x)} dx}{\int e^{\theta^T s(x)} dx} \tag{51}$$

$$= \int s(x) e^{\theta^T s(x) - \log Z_\theta} \mathrm{d}x \tag{52}$$

$$= \mathbb{E}_\theta[s(x)]; \tag{53}$$

and

$$H_\theta(\log Z_\theta) = \int s(x) s(x)^T e^{\theta^T s(x) - \log Z_\theta} \mathrm{d}x - \left( \int s(x) e^{\theta^T s(x) - \log Z_\theta} \mathrm{d}x \right) (\nabla_\theta \log Z_\theta)^T \tag{54}$$

$$= \int s(x) s(x)^T e^{\theta^T s(x) - \log Z_\theta} \mathrm{d}x \tag{55}$$

$$- \left( \int s(x) e^{\theta^T s(x) - \log Z_\theta} \mathrm{d}x \right) \left( \int s(x) e^{\theta^T s(x) - \log Z_\theta} \mathrm{d}x \right)^T \tag{56}$$

$$= \mathbb{E}_\theta[s(x) s(x)^T] - \mathbb{E}_\theta[s(x)] \mathbb{E}_\theta[s(x)]^T \tag{57}$$

$$= \mathbb{V}_\theta[s(x)]. \tag{58}$$

Using the Hessian of $\log Z_\theta$, we can express directly the Hessian of the log-likelihood $\ell(\theta)$:

$$H(\ell(\theta)) = -\mathbb{V}_\theta[s(x)]. \tag{59}$$

The covariance matrix $\mathbb{V}_\theta[s(x)]$ is positive semi-definite thus the hessian $H(\ell(\theta))$ is negative semi-definite. Hence, $\ell(\theta)$ is concave. $\qquad \square$

### A.4   Theorem 3.1: Concavity of SNL in exponential families

**Theorem 3.1.** *If $(p_\theta)_\theta$ is a canonical exponential family, then $\ell_{\mathrm{SNL}}(\theta, b)$ is jointly concave.*

*Proof.* Using the same notations as the previous proof, our exponential family is parameterised as $p_\theta(x) = e^{\theta^T s(x) - \log Z_\theta}$, where $s(x)$ is the sufficient statistics and $\theta$ the natural parameters. We again assume without loss of generality that we observe a single data point $x \in \mathcal{X}$.
The self-normalised log-likelihood is as follows:

$$\ell_{\mathrm{SNL}}(\theta, b) = \theta^T s(x) - b - e^{-b + \log Z_\theta} + 1 \tag{60}$$

$$= \theta^T s(x) - b + 1 - \int e^{\theta^T s(x) - b} \mathrm{d}x. \tag{61}$$

Since the first term of the equation is affine, we will show that the function $(\theta, b) \mapsto e^{-b + \log Z_\theta}$ is jointly convex in $(\theta, b)$. Let $(b_1, \theta_1)$ and $(b_2, \theta_2)$ any given pair of parameters and let $\lambda \in [0, 1]$:

$$\int e^{(\lambda\theta_1 + (1-\lambda)\theta_2)^T s(x) - (\lambda b_1 + (1-\lambda)b_2)} \mathrm{d}x = \int e^{\lambda(\theta_1^T s(x) - b_1) + (1-\lambda)(\theta_2^T s(x) - b_2)} \mathrm{d}x \tag{62}$$

$$\geq \left[ \int \left( \lambda e^{\theta_1^T s(x) - b_1} + (1-\lambda)e^{\theta_2^T s(x) - b_2} \right) \mathrm{d}x \right] \tag{63}$$

$$= \lambda \int e^{\theta_1^T s(x) - b_1} \mathrm{d}x + (1-\lambda) \int e^{\theta_2^T s(x) - b_2} \mathrm{d}x. \tag{64}$$

The function $(\theta, b) \mapsto e^{-b + \log Z_\theta}$ is convex jointly in $(\theta, b)$, thus $(\theta, b) \mapsto \ell_{\mathrm{SNL}}(\theta, b)$ is also convex jointly in $(\theta, b)$ which concludes the proof. $\qquad\square$

## B    The Gaussian case

We consider a univariate Gaussian with unknown mean $\theta \in \mathbb{R}$ and known unit variance. The model is parameterised as an exponential family with energy and normalising constant:

$$E_\theta(x) = -\theta x, \; \log Z_\theta = \frac{1}{2}\theta^2, \tag{65}$$

the base measure being the standard Gaussian measure.
For a dataset $(x_1, ..., x_n) \in \mathbb{R}^n$, the log-likelihood is:

$$\ell(\theta) = \frac{1}{n} \sum_{i=1}^{n} x_i \theta - \frac{1}{2}\theta^2, \tag{66}$$

which is concave and is maximised at $\hat{\theta}_{\mathrm{ML}} = \bar{x}_n$. The SNL equals:

$$\ell_{\mathrm{SNL}}(\theta, b) = \frac{1}{n} \sum_{i=1}^{n} x_i \theta - b - Z_\theta e^{-b} + 1 \tag{67}$$

$$= \frac{1}{n} \sum_{i=1}^{n} x_i \theta - b - e^{\frac{1}{2}\theta^2 - b} + 1, \tag{68}$$

which is also concave and is maximised at $(\hat{\theta}_{\mathrm{SNL}}, \hat{b}_{\mathrm{SNL}}) = (\bar{x}_n, \bar{x}_n^2/2)$.

## C    The Bernoulli case

In the same vein as in Appendix B, we derive here the SNL for a Bernoulli distribution, in order to gain basic insights. We consider a Bernoulli distribution with unknown natural parameter $\theta \in \mathbb{R}$ ($\theta$ here is the logit of the probability of success). The model is parameterised as an exponential family with energy and normalising constant:

$$E_\theta(x) = -\theta x, \; \log Z_\theta = \log\left(1 + e^\theta\right), \tag{69}$$

the base measure being the uniform measure on $\{0, 1\}$.
For a dataset $(x_1, ..., x_n) \in \{0, 1\}^n$, the log-likelihood is:

$$\ell(\theta) = \frac{1}{n} \sum_{i=1}^{n} x_i \theta - \log\left(1 + e^\theta\right), \tag{70}$$

which is concave and is maximised at $\hat{\theta}_{\mathrm{ML}} = \mathrm{logit}(\bar{x}_n)$. The SNL equals:

$$\ell_{\mathrm{SNL}}(\theta, b) = \frac{1}{n} \sum_{i=1}^{n} x_i \theta - b - Z_\theta e^{-b} + 1 \tag{71}$$

$$= \frac{1}{n} \sum_{i=1}^{n} x_i \theta - b - e^{-b}\left(1 + e^\theta\right) + 1, \tag{72}$$

which is also concave and is maximised at $(\hat{\theta}_{\mathrm{SNL}}, \hat{b}_{\mathrm{SNL}}) = (\mathrm{logit}(\bar{x}_n), \log(1 + e^{\mathrm{logit}(\bar{x}_n)}))$.

# D  The Kullback-Leibler divergence for un-normalised densities

We consider a measured space $\mathcal{X}$, equipped with a base measure $\mathrm{d}x$ (typically the Lebesgue or the counting measure). Let $f_1, f_2$ be the densities of two finite measures. The Kullback-Leibler divergence between these is then defined as

$$\mathrm{KL}(f_1||f_2) = \int \log\left(\frac{f_1(x)}{f_2(x)}\right) f_1(x)\mathrm{d}x + \left(\int f_2(x)\mathrm{d}x - \int f_1(x)\mathrm{d}x\right). \tag{73}$$

It is clear that this reduces to the usual KL when $f_1$ and $f_2$ are probability densities. For more details, see, for instance, Amari & Nagaoka (2000, Section 3.6) or Stummer & Vajda (2010).

Why is this a sensible generalisation? We can write our un-normalised densities as $f_1 = \mu_1 p_1$ and $f_2 = \mu_1 p_2$, where

$$\mu_1 = \int f_1(x)\mathrm{d}x, \ \mu_2 = \int f_2(x)\mathrm{d}x. \tag{74}$$

Plugging this into equation 73 gives

$$\mathrm{KL}(f_1||f_2) = \mu_1 \mathrm{KL}(p_1||p_2) + \mu_1 \log\left(\frac{\mu_1}{\mu_2}\right) + (\mu_2 - \mu_1) \tag{75}$$

$$= \mu_1 \left(\mathrm{KL}(p_1||p_2) + h\left(\frac{\mu_2}{\mu_1}\right)\right), \tag{76}$$

where $h : t \mapsto t - 1 - \log t$. Since $h(t) > 0$ for all $t \neq 1$ and $h(1) = 0$, we will have

- $\mathrm{KL}(f_1||f_2) \geq 0$

- $\mathrm{KL}(f_1||f_2) = 0$ if and only if $f_1 = f_2$.

This means that this generalised KL enjoys some of the nice properties of the usual KL, which motivates its use for statistical inference.

Another interesting property that is a direct consequence of Eq. (75) is that

$$\mathrm{KL}(p_1||p_2) = \min_{c>0} \mathrm{KL}(p_1||cf_2), \tag{77}$$

which means that we can recover the KL between probability densities by minimising the KL between un-normalised densities, transforming the computation of the normalising constant into an optimisation problem. This justifies Eq. (19). Another interpretation of this property is that the KL between $p_1$ and the set $\{cf_2; c > 0\}$ is just the KL between $p_1$ and $p_2$.

The KL divergence between two un-normalised densities relates to the self-normalised log-likelihood as such:

$$\mathrm{KL}(p_{\mathrm{data}}||ce^{-E_\theta}) = \int \log\left(\frac{p_{\mathrm{data}}(x)}{ce^{-E_\theta(x)}}\right) p_{\mathrm{data}}(x)(x)\mathrm{d}x + cZ_\theta - 1 \tag{78}$$

$$= -\int \left(e^{-E_\theta(x)}p_{\mathrm{data}}(x) - \log c + cZ_\theta - 1\right)\mathrm{d}x + \underbrace{\int \log(p_{\mathrm{data}}(x))p_{\mathrm{data}}(x)\mathrm{d}x}_{\text{does not depend on } \theta \text{ nor } c}. \tag{79}$$

As we assume we have access to an i.i.d. dataset $x_1, ..., x_n$, we can estimate the above quantity:

$$\mathrm{KL}(p_{\mathrm{data}}||ce^{-E_\theta}) \approx -\frac{1}{n}\sum_{i=1}^{n} e^{-E_\theta(x_i)} - \log c + cZ_\theta - 1 + \int \log(p_{\mathrm{data}}(x))p_{\mathrm{data}}(x)\,\mathrm{d}x \tag{80}$$

$$= -\ell_{\mathrm{SNL}}(\theta, \log c) + \int \log(p_{\mathrm{data}}(x))p_{\mathrm{data}}(x)\,\mathrm{d}x. \tag{81}$$

This implies that maximising the self-normalised log-likelihood will, asymptotically, resemble minimising the generalised Kullback-Leibler divergence.

# E Link with the Donsker-Varadhan representation

## E.1 The Donsker-Varadhan representation

The Donsker-Varadhan representation (Donsker & Varadhan, 1975) provides a variational (Fenchel dual) formulation of the Kullback-Leibler divergence between two distributions $\mathbb{P}$ and $\mathbb{B}$:

$$\mathrm{KL}(\mathbb{P}\|\mathbb{B}) = \sup_{h \in C_b^0(\mathbb{R}^d)} \left\{ 1 + \int h \, d\mathbb{P} - \int e^h \, d\mathbb{B} \right\}, \tag{82}$$

where $C_b^0(\mathbb{R}^d)$ denotes the set of continuous bounded functions from $\mathbb{R}^d$ to $\mathbb{R}$. The supremum is attained when $h = \log(d\mathbb{P}/d\mathbb{B})$, i.e., the log-density ratio between the two distributions.

This variational formulation has inspired several approaches to training energy-based models and defining divergences between distributions. Different methods arise from different choices of parametrisation or constraints the function $h$. In this section, we describe how SNL can be derived from this formulation and contrast it with two related approaches: Generalized Energy Based Models (Arbel et al., 2020) and KALE Flow (Glaser et al., 2021).

## E.2 From Donsker-Varadhan to SNL

To connect with the Donsker-Varadhan representation, we take $\mathbb{P} = p_{\mathrm{data}}$ (the data distribution) and define the base measure $d\mathbb{B}$ as the Lebesgue measure directly. Crucially, SNL does not optimise over all functions $h$; instead we restrict the function $h$ to be of the form $h(x) = -E_\theta(x) - b$ where $E_\theta$ is an energy function parameterised by $\theta$ and $b \in \mathbb{R}$ is a scalar. Substituting this into Eq. (82) yields:

$$1 + \int h \, d\mathbb{P} - \int e^h \, d\mathbb{B} = 1 - \int E_\theta(x) \, d\mathbb{P}(x) - b - e^{-b} \int e^{-E_\theta(x)} \, d\mathbb{B}(x) \tag{83}$$

$$= 1 - \int E_\theta(x) \, d\mathbb{P}(x) - b - e^{-b} Z_\theta. \tag{84}$$

where $Z_\theta = \int e^{-E_\theta(x)} \, d\mathbb{B}(x)$ is the partition function of the energy-based model defined by $E_\theta$. In our case, $E_\theta$ is parameterized by a neural network $f_\theta$. When $\mathbb{P}$ is the empirical data distribution, this becomes exactly the SNL objective (Eq. (10) in the main text):

$$\ell_{\mathrm{SNL}}(\theta, b) = \frac{1}{n} \sum_{i=1}^{n} \left[ -E_\theta(x_i) - b - e^{-b} Z_\theta + 1 \right]. \tag{85}$$

**Using a base distribution.** The above derivation assumes that the function $E_\theta$ is defined with respect to the Lebesgue measure, which is not the case in the general setting when parameterized by an arbitrary neural network $f_\theta$. To alleviate this issue, we introduced in Section 2.4 a fixed base density $d(x)$ and define the energy-based model as $E_\theta(x) = f_\theta(x) - \log d(x)$.

From the Donsker-Varadhan perspective, this can be interpreted in two ways:

- We can take the base measure $\mathbb{B}$ to be the measure with density $d(x)$ with respect to the Lebesgue measure $dx$, i.e. $d\mathbb{B}(x) = d(x)dx$. In this case, the complete energy function $E_\theta$ is still defined as $E_\theta(x) = f_\theta(x)$. The normalisation constant becomes $Z_\theta = \int e^{-f_\theta(x)} d\mathbb{B}(x) = \int e^{-f_\theta(x)} d(x) \, dx$.

- Alternatively, we can take the base measure $\mathbb{B}$ to be the Lebesgue measure, and define the energy function as $E_\theta(x) = f_\theta(x) - \log d(x)$. In this case, the SNL objective is obtained by substituting $h(x) = -E_\theta(x) + \log d(x) - b$ into the Donsker-Varadhan representation, and the base density $d(x)$ appears as a multiplicative factor in the partition function $Z_\theta = \int e^{-f_\theta(x)} d(x) \, d\mathbb{B}(x) = \int e^{-f_\theta(x)} d(x) \, dx$.

In both cases, we can use a proposal distribution $q$ to obtain unbiased estimates of the partition function and its gradients, without affecting the population-level objective or the correspondence with maximum likelihood. In particular, if one chooses proposal density $q(x)$ to be the base density $d(x)$, we obtain a special case of SNL loss that corresponds to a special case of the GEBM.

### E.3 On the difference with Generalized Energy Based Models

The Generalized Energy Based Model (GEBM) is derived by using a slightly changed parameterisation. Instead, of having $\mathbb{B}$ as the Lebesgue measure, the base measure $\mathbb{B}$ is a learnable distribution, typically an implicit generative model such as a GAN generator. The GEBM objective is obtained by substituting $h(x) = -E(x) - b$ into the Donsker-Varadhan representation with this learnable base $\mathbb{B}$:

$$\text{GEBM}(E, b, \mathbb{B}) = 1 - \int E(x)\, d\mathbb{P}(x) - b - e^{-b} \int e^{-E(x)}\, d\mathbb{B}(x). \tag{86}$$

In the general case, this base measure does not admit a density with respect to the Lebesgue measure, and the energy function $E$ is only defined with respect to this base measure.

**No separation between base and proposal.** A key distinction between SNL and GEBM lies in the relationship between the base measure and the proposal distribution. In SNL, the base measure $d(x)$ is part of the model definition, while the proposal $q(x)$ used for importance sampling can be chosen independently—the population-level objective is invariant to this choice. In GEBM, no such separation exists: samples for estimating the partition function are drawn directly from the learned base $\mathbb{B}$ itself. The base and proposal are thus identical, and both change during training. This coupling means that changing $\mathbb{B}$ simultaneously changes both the probabilistic model and the distribution from which samples are drawn.

In the special case where the base distribution $\mathbb{B}$ admits a density with respect to the Lebesgue measure, the GEBM objective can be understood as SNL where the base distribution and the proposal distribution are forced to coincide. However, in the general case where $\mathbb{B}$ is an implicit model (e.g., a GAN generator), this interpretation does not apply, and the lack of separation between base and proposal becomes a fundamental characteristic of the method.

### E.4 On the difference with KALE Flow

KALE Flow (Glaser et al., 2021) also builds on the Donsker-Varadhan representation, but takes a fundamentally different approach. Rather than evaluating the dual at a fixed parametrisation, KALE *optimises* over a restricted function class $\mathcal{H}$ and introduces a regularisation term:

$$\text{KALE}_\alpha(\mathbb{P}\|\mathbb{B}) = (1 + \alpha) \sup_{h \in \mathcal{H}} \left\{ 1 + \int h\, d\mathbb{P} - \int e^h\, d\mathbb{B} - \frac{\alpha}{2}\|h\|_{\mathcal{H}}^2 \right\}, \tag{87}$$

where $\mathcal{H}$ is typically a reproducing kernel Hilbert space (RKHS) and $\alpha > 0$ is a regularisation parameter.

The KALE objective defines a surrogate divergence between $\mathbb{P}$ and $\mathbb{B}$ that depends on both the function class $\mathcal{H}$ and the regularisation strength $\alpha$. For finite $\alpha$ and a restricted $\mathcal{H}$, the minimiser of KALE does not, in general, coincide with the maximum likelihood solution. Instead, KALE is designed as a tool for defining gradient flows: the optimal $h^*$ in Eq. (87) provides a transport direction that moves samples from $\mathbb{B}$ toward $\mathbb{P}$. This makes KALE well-suited for particle-based inference and sampling methods, where the goal is to iteratively transport a source distribution toward a target.

The key distinction with SNL is that KALE optimises over functions $h$ to define a divergence, whereas SNL fixes the parametrisation $h = -E_\theta - b$ and optimises over $\theta$ and $b$ to perform maximum likelihood estimation. In SNL, the function $h$ is not a transport potential but rather encodes the energy-based model whose parameters we wish to learn. As a result, SNL retains an exact correspondence with maximum likelihood, while KALE defines a relaxed divergence useful for gradient flows.

## F Algorithms

---

**Algorithm 1:** Training an EBM for density estimation using SNL loss and proposal $q$.

---

**input** : Learning iterations, $T$; learning rate, $\eta$; initial parameters, $\{\theta_0, b_0\}$; observed examples, $\{x_i\}_{i=1}^n$; batch size, $n_b$; number of samples from the proposal $q$, $M$.

**output**: $\theta_T, b_T$.

**for** $t = 0 : T - 1$ **do**

    1. **Mini-batch**: Sample observed examples $\{x_i\}_{i=1}^{n_b}$.
    2. **Proposal sampling**: Sample M elements from the proposal $x_m \sim \tilde{q}(x_m)$
    3. **Learn EBM parameters** $\theta$: Update $\theta_{t+1} = \theta_t - \eta \hat{\nabla}_\theta \ell_{\text{SNL}}(\theta, b)$ using $\hat{\nabla}_\theta \ell_{\text{SNL}}(\theta_t, b)$ defined in Eq. (15).
    4. **Learn** $b$: Update $b_{t+1} = b_t - \eta \hat{\nabla}_b \ell_{\text{SNL}}(\theta, b_t)$ using $\hat{\nabla}_b \ell_{\text{SNL}}(\theta, b_t)$ in defined Eq. (16).

---

---

**Algorithm 2:** Training a VAE with EBM prior using the SNELBO loss.

---

**input** : Learning iterations, $T$; learning rate, $\eta$; initial parameters, $\{\theta_0, \gamma_0, \phi_0, b_0\}$; observed examples, $\{x_i\}_{i=1}^{n}$; batch size, $n_b$; number of samples from the base $d$, $M$.

**output:** $\theta_T, \gamma_T, \phi_T, b_T$.

**for** $t = 0 : T - 1$ **do**

    1. **Mini-batch**: Sample observed examples $\{x_i\}_{i=1}^{n_b}$.

    2. **Proposal sampling**: Sample M elements from the base $x_m \sim \tilde{d}(x_m)$

    3. **Learn EBM parameters** $\theta$: Update $\theta_{t+1} = \theta_t - \eta\hat{\nabla}_\theta\mathcal{L}_{\text{SNL}}((\theta, \gamma, \phi, b)$ using Eq. (28).

    4. **Learn VAE parameters**: Update $\{\gamma, \phi\}_{t+1} = \{\gamma, \phi\}_t - \eta\hat{\nabla}_{\{\gamma,\phi\}}\mathcal{L}_{\text{SNL}}(\theta, \gamma, \phi, b)$ using Eq. (28).

    5. **Learn** $b$: Update $b_{t+1} = b_t - \eta\hat{\nabla}_b\ell_{\text{SNL}}(\theta, b_t)$ using Eq. (28).

---

# G  Derivation of the SNELBO

Using the variational distribution $q_\gamma(z|x)$, we can write the regular ELBO for the VAE with the energy-based prior as

$$\mathcal{L}(\theta, \phi, \gamma) = \mathbb{E}_{q_\gamma(z|x)}[\log p_\phi(x|z)] + \mathbb{E}_{q_\gamma(z|x)}\left[\log \frac{e^{-E_\theta(z)}d(z)}{q_\gamma(z|x)Z_\theta}\right], \tag{88}$$

which is a lower bound, $\ell(\theta, \phi) \geq \mathcal{L}(\theta, \phi, \gamma)$, on the log-likelihood

$$\ell(\theta, \phi) = p_{\theta,\phi}(x) = \int p_\phi(x|z)p_\theta(z)\, \mathrm{d}z, \tag{89}$$

where we left out the sum over data to simply the notation. Using Lemma 2.1, we define the SNELBO as

$$\mathcal{L}_{\text{SNL}}(\theta, \phi, \gamma, b) = \mathbb{E}_{q_\gamma(z|x)}[\log p_\phi(x|z)] + \mathbb{E}_{q_\gamma(z|x)}\left[\log \frac{d(z)}{q_\gamma(z|x)}\right]$$
$$+ \mathbb{E}_{q_\gamma(z|x)}\left[-E_\theta(z) - b\right] - Z_\theta e^{-b} + 1, \tag{90}$$

which can be written using the base distribution $d$,

$$\mathcal{L}_{\text{SNL}}(\theta, \phi, \gamma, b) = \mathbb{E}_{q_\gamma(z|x)}\left[\log p_\phi(x|z)\right] + \mathbb{E}_{q_\gamma(z|x)}\left[\log \frac{d(z)}{q_\gamma(z|x)}\right]$$
$$+ \mathbb{E}_{q_\gamma(z|x)}\left[-E_\theta(z) - b\right] - \mathbb{E}_{d(z)}\left[e^{-E_\theta(z)-b}\right] + 1 \tag{91}$$

Lemma 2.1 gives directly the following results :

$$\ell(\theta, \phi) \geq \mathcal{L}(\theta, \phi, \gamma) \geq \mathcal{L}_{\text{SNL}}(\theta, \phi, \gamma, b) \tag{92}$$

# H  Regression datasets

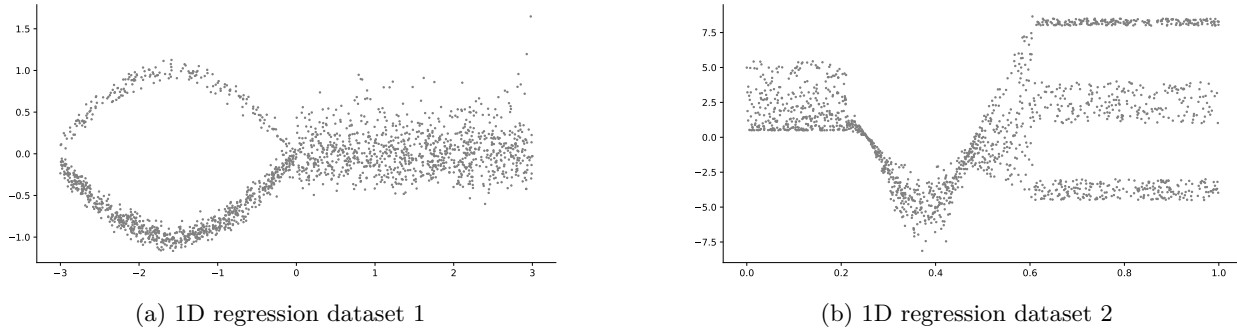

(a) 1D regression dataset 1          (b) 1D regression dataset 2

Figure 5: Visualisation of the two toy regression datasets. The $x$-axis corresponds to the input of the regression ($x$ in Eq. (27)) and the $y$-axis corresponds to the regressed value ($y$ in Eq. (27)).

The first dataset set, on the left-hand side of Fig. 5, is a mixture of two Gaussians with weights 0.2 and 0.8 for negative values on the $x$-axis and a log-normal distribution $\text{Log} -\mathcal{N}(0., 0.25)$ for positive values of $x$. There are 2000 training samples that we generated by uniformly sampling values in $[-3, 3]$.

The second dataset, on the right-hand side of Fig. 5, is defined for $x$ in $[0, 1]$ and is divided into four chunks. The first one, for $x < 0.21$, is sampled from $\text{Beta}(\alpha = 0.5, \beta = 1)$; the second one, for $0.21 \leq x < 0.47$ is sampled from $\mathcal{N}\left(\mu = 3 \cdot \cos x - 2, \sigma = |3 \cdot \cos x - 2|\right)$; the third one for $0.47 \leq x < 0.61$ from an increasing uniform distribution; the fourth and last one, for $0.61 \leq x \leq 1$ is obtained from a mixture of uniform distribution, $\mathcal{U}(8, 0.5), \mathcal{U}(1, 3)$ and $\mathcal{U}(-4.5, 1.5)$.

# I Experimental setting

## I.1 Multivariate von Mises

The multivariate von Mises distribution has a density of the following form:

$$\mathcal{M}v\mathcal{M}(x) = \frac{1}{Z_{\theta, \mathbf{\Lambda}, \boldsymbol{\kappa}}} \exp\left(\boldsymbol{\kappa}^T c(x, \theta) + s(x, \theta)^T \mathbf{\Lambda} s(x, \theta)\right), \tag{93}$$

with $Z_{\theta, \mathbf{\Lambda}, \boldsymbol{\kappa}}$ an unknown normalisation constant, $\theta$ the localisation parameter, $\mathbf{\Lambda}$ the correlation parameter and $\boldsymbol{\kappa}$ the concentration parameters verifying: $-\pi < \theta_i \leq \pi, -\pi < \mu_i \leq \pi, \boldsymbol{\kappa}_i \geq 0, -\infty < \lambda_{ij} < \infty$ and $\mathbf{\Lambda}$ is such that $(\mathbf{\Lambda})_{ij} = \lambda_{ij} = \lambda_{ji}, \quad i \neq j, \lambda_{ii} = 0$. We defined $c(x, \theta)^T = \left(\cos(x - \theta_1), \cos(x - \theta_2), \ldots, \cos(x - \theta_p)\right)$ and $s(x, \theta)^T = \left(\sin(x - \theta_1), \sin(x - \theta_2), \ldots, \sin(x - \theta_p)\right)$.

### I.1.1 Implementation of the Multivariate von Mises

To implement the multivariate von Mises (mVM) distribution, we adopt a parameterisation that ensures the required constraints on the concentration and interaction parameters:

- **Concentration parameters $\boldsymbol{\kappa}$** are constrained to be non-negative. We enforce this by parameterising in the log-domain and applying the exponential function:

$$\kappa_i = \exp(\eta_i), \quad \text{where } \eta_i \in \mathbb{R}.$$

- **Interaction matrix** $\Lambda$ must be symmetric to ensure valid dependencies among angular variables. We construct it by symmetrising a lower-triangular matrix:

$$\Lambda = L + L^\top,$$

where $L$ is a lower-triangular matrix with unconstrained entries.

### I.1.2 Sampling the dataset from $\mathcal{M}v\mathcal{M}$ using Gibbs Sampling

This distribution has an unknown normalisation constant, which prevents the use of the inverse cumulative distribution function for sampling. In fact, there is no direct way of sampling from this distribution. However, the distribution of a single dimension conditioned on the other leads to a simple von Mises distribution $v\mathcal{M}(\mu_{j\text{-rest}}, \boldsymbol{\kappa}_{j\text{-rest}})$ with the following parameters:

$$\mu_{j\text{-rest}}^{(i)} = \mu_j + \tan^{-1}\left\{\left[\sum_{\ell \neq j} \lambda_{j\ell} \sin(x_{\ell i} - \theta_\ell)\right] / \boldsymbol{\kappa}_j\right\} \tag{94}$$

$$\boldsymbol{\kappa}_{j\text{-rest}}^{(i)} = \left\{\boldsymbol{\kappa}_j^2 + \left[\sum_{\ell \neq j} \lambda_{j\ell} \sin(x_{\ell i} - \theta_\ell)\right]^2\right\}^{1/2} \tag{95}$$

Sampling from a standard univariate Von Mises distribution Best & Fisher (1979) is possible. Thus, by successively updating the conditional parameters $(\mu_{j\text{-rest}}, \boldsymbol{\kappa}_{j\text{-rest}})$ and sampling the corresponding coordinate from the resulting von Mises distribution using Gibbs sampling, it is possible to sample from the multivariate von Mises distribution.

Following the experiments in Mardia et al. (2008), we consider four parameter groups for the Multivariate von Mises distribution that encompass different scenarios. These parameters can be found in Table 17. For each group of parameters, we sample 100 data points by repeating 50 Gibbs sampling cycles over each dataset. We fix the location parameters to 0 and assume they are known.

### I.1.3 Training Hyperparameters

For each set of parameters, we simulate five datasets of 100 samples with Gibbs sampling, where each sample is separated by 50 steps of filtering (following the setup in Mardia et al. (2008)). We report the average parameter estimate and standard deviation for parameters estimated with SNL in Fig. 2 and Table 17. We trained MvM with SNL for 10 000 steps using 100 samples from a Uniform Proposal on the Torus using the Adam optimiser with a learning rate of 0.1.

### I.2 Density estimation with Truncated distribution

We are interested in obtaining the parameters of the constrained mixture of Gaussians in Fig. 3.

### I.2.1 Mixture parameterisation

We parametrise the multivariate Gaussian distributions to ensure the covariance matrix is symmetric and positive definite:

- **Diagonal elements** of the covariance matrix are parameterised using a log-scale transformation:

$$\Sigma_{\text{diag}} = \exp(\boldsymbol{\sigma}),$$

  where $\boldsymbol{\sigma} \in \mathbb{R}^d$ is a vector of unconstrained parameters.

- **Off-diagonal structure** is captured via a Cholesky decomposition $LL^\top$, where $L$ s lower triangular.

- **The full covariance** of a single Gaussian of the mixture is obtained as follows :

$$\Sigma = \Sigma_{diag} + LL^\top, \quad \text{where } L \text{ is lower triangular.}$$

We train a mixture of $K = 3$ full-covariance Gaussian:

$$p(x) = \sum_{k=1}^{K} \pi_k \, \mathcal{N}(x \mid \mu_k, \Sigma_k), \tag{96}$$

which is then truncated with a known truncation $h : \mathcal{X} \to \{0, 1\}$:

$$p_{\text{Truncated}}(x) \propto e^{-\mathbf{1}_{h(x)=1} \sum_{k=1}^{K} \mathcal{N}(x|\mu_k, \Sigma_k) + \mathbf{1}_{h(x)=0} C}. \tag{97}$$

In practice, we use $C = 1e8$ to enforce high energy outside the truncated domain.

We train the model using SNL with a simple univariate Gaussian Distribution as proposal. We optimise the model using Adam Kingma & Ba (2015) for 10 000 steps and learning rate 0.01, and we chose $C = 1e9$ to mimic a potential barrier. The parameters are initialised using standard Gaussians and K-Means centres.

### I.3 Density estimation

For density estimation, we parametrise the energy $E_\theta$ using a neural network with parameters given in Table 6.

| $E_\theta$ | Activation | Output shape |
|---|---|---|
| Fully Connected | ReLU | $2 \times 200$ |
| Fully Connected | ReLU | $200 \times 100$ |
| Fully Connected | ReLU | $100 \times 50$ |
| Fully Connected | ReLU | $50 \times 50$ |
| Fully Connected | ReLU | $50 \times 1$ |
| Total trainable parameters | | 30450 |

Table 6: $E_\theta$ for the toy distribution estimation

### I.4   Energy Based Regression

In Energy Based Regression, we consider an architecture similar to Gustafsson et al. (2020) in which a feature extractor $h_\theta$ is parameterised as a neural network. The outputs of the feature extractor are fed to both the energy $E_\theta(x, y) = f_\theta(h_\theta(x), y)$ with overhead $f_\theta$ parameterised by a neural network. The normalisation constant $Z_x$ is obtained by considering a neural network overhead over the feature extractor $b_\phi(h(x))$.

When indicated, we use a Mixture Density Network as proposal distribution. A Mixture Density Network (MDN) models a conditional distribution $q(y \mid x)$ using a mixture of Gaussians, where the mixture parameters are predicted by a neural network conditioned on $x$. That is

$$q(y \mid x) = \sum_{k=1}^{K} \pi_k(x) \mathcal{N}(y \mid \mu_k(x), \sigma_k(x)) \tag{98}$$

where:

- $\pi_k(x)$ are the mixing coefficients, satisfying $\pi_k(x) \geq 0$ and $\sum_{k=1}^{K} \pi_k(x) = 1$,

- $\boldsymbol{\mu}_k(x) \in \mathbb{R}^d$ is the mean of the $k$-th component,

- $\sigma_k(x) \in \mathbb{R}^d$ is the standard deviation of the $k$-th component.

All parameters $\{\pi_k, \boldsymbol{\mu}_k, \boldsymbol{\Sigma}_k\}_{k=1}^{K}$ are predicted by a neural network that use the extracted features $h(x)$ as input.

#### I.4.1   Toy regression

For Toy regression, we follow the same training procedure as Gustafsson et al. (2022) but replacing the NCE loss with the SNL loss. As such, the model is trained for 75 epochs, a batch size $B = 32$, $M = 1024$ samples and a learning rate 0.001.

For toy regression problems defined in Fig. 5 with results in Table 20 and Table 3, we parametrise the feature extractor $h_\theta$ as Table 7, the energy overhead as Table 8 and the normalisation constant as Table 10. The mixture density network's parameters are given in Table 9.

| Feature extractor | Activation | Output shape |
|---|---|---|
| Fully Connected | ReLU | $10 \times 10$ |
| Fully Connected | ReLU | $10 \times 10$ |
| Fully Connected | ReLU | $10 \times 1$ |
| Total trainable parameters | | 210 |

Table 7:  Feature extractor. Inputs $x$ and outputs $h_x$

| $E_\theta$ | Activation | Output shape |
|---|---|---|
| Input $y \to$ Output $f(y)$ | | |
| Fully Connected | ReLU | $1 \times 16$ |
| Fully Connected | ReLU | $16 \times 32$ |
| Fully Connected | ReLU | $32 \times 64$ |
| Fully Connected | ReLU | $64 \times 128$ |
| Concatenation of $h_x$ and $f(y)$ | | |
| Fully Connected | ReLU | $144 \times 10$ |
| Fully Connected | ReLU | $10 \times 1$ |
| Total trainable parameters | | 30450 |

Table 8: $E_\theta$ for 1d regression estimation.

| MDN | Activation | Output shape |
|---|---|---|
| Input $h_x$ | | |
| Fully Connected | ReLU | $10 \times 10$ |
| Fully Connected | ReLU | $10 \times K$ |
| Total trainable parameters | | $100 + 10 \times K$ |

Table 9: Neural network estimating one parameter of the MDN with $K$ components in the mixture.

| $b_\phi$ | Activation | Output shape |
|---|---|---|
| Input $h_x$ | | |
| Fully Connected | ReLU | $10 \times 10$ |
| Fully Connected | ReLU | $10 \times 1$ |
| Total trainable parameters | | 110 |

Table 10: Neural network estimating the normalization constant $Z_{\theta,x}$ for every $x$.

### I.4.2 Image for regression

For image regression, we follow the same training procedure as Gustafsson et al. (2022). Each model is run for 75 epochs, with batch size 32, learning rate 0.001 and $M = 64$ (unless specified otherwise) samples from the proposal distribution.

For image regression problems with results in Table 4 and Table 21, the feature extractor is a Resnet-18 He et al. (2016) from torchvision Paszke et al. (2019). The energy overhead is parameterised as Table 11 and the normalisation constant as Table 13. The mixture density network's parameters are given in Table 12.

| $E_\theta$ | Activation | Output shape |
|---|---|---|
| Input $y \to$ Output $f(y)$ | | |
| Fully Connected | ReLU | $1 \times 16$ |
| Fully Connected | ReLU | $16 \times 32$ |
| Fully Connected | ReLU | $32 \times 64$ |
| Fully Connected | ReLU | $64 \times 128$ |
| Concatenation of $h_x$ and $f(y)$ | | |
| Fully Connected | ReLU | $640 \times 640$ |
| Fully Connected | ReLU | $640 \times 1$ |
| Total trainable parameters | | 420816 |

Table 11: $E_\theta$ for 1d regression estimation.

| MDN | Activation | Output shape |
|---|---|---|
| Input $h_x$ | | |
| Fully Connected | ReLU | $512 \times 512$ |
| Fully Connected | ReLU | $512 \times K$ |
| Total trainable parameters | | $262144 + 512 \times K$ |

Table 12: Neural network estimating one parameter of the MDN with $K$ components in the mixture. We use three such networks for $\pi_\psi, \mu_\psi, \sigma_\psi$.

| EBM Model for BinaryMNIST | | |
|---|---|---|
| Layers | In-Out Size | Stride |
| Input: $z$ | 100 | |
| Linear, LReLU | 200 | - |
| Linear, LReLU | 200 | - |
| Linear | 1 | - |

Table 14: Architecture of the Energy-Based Model (EBM) prior used for all the datasets in the VAE with EBM prior.

| $b_\phi$ | Activation | Output shape |
|---|---|---|
| Input $h_x$ | | |
| Fully Connected | ReLU | $512 \times 512$ |
| Fully Connected | ReLU | $512 \times 1$ |
| Total trainable parameters | | 262656 |

Table 13: Neural network estimating the normalization constant $Z_{\theta,x}$ for every $x$.

### I.4.3 Training the proposal distribution for MDN

Following the method from Gustafsson et al. (2022), the MDN proposal is trained by minimising a sum of the negative log-likelihood and the KL divergence with the EBM:

$$\ell_\psi = \frac{1}{N}\sum_{i=1}^{N}\frac{1}{2}\log\left(\frac{1}{M}\sum_{m=1}^{M}\frac{e^{-E_\theta\left(x_i,y_i^{(m)}\right)}}{q_\psi\left(y_i^{(m)}\mid x_i;\phi\right)}\right) - \frac{1}{2}\log q_\psi\left(y_i\mid x_i;\phi\right) \tag{99}$$

This allows us to guide the proposal distribution towards the EBM. While this is trained at the same time as the EBM, only the head of the MDN is updated, while the feature extractor $h(x)$ remains fixed.

### I.5 VAE with prior EBM

All the parameters of the VAE with prior EBM are given in Table 14, Table 15 and **??**.

| Generator Model for BinaryMNIST, ngf = 16 | | |
|---|---|---|
| Layers | In-Out Size | Stride |
| Input: $z$ | 16×1×1 | |
| 4×4 convT(ngf × 8), LReLU | 4×4×(ngf × 8) | 1 |
| 3×3 convT(ngf × 4), LReLU | 7×7×(ngf × 4) | 2 |
| 4×4 convT(ngf × 2), LReLU | 14×14×(ngf × 2) | 2 |
| 4×4 convT(1), Sigmoid | 28×28×1 | 2 |
| Generator Model for CIFAR-10, ngf = 128 | | |
| Input: $z$ | 1×1×128 | |
| 8×8 convT(ngf × 8), LReLU | 8×8×(ngf × 8) | 1 |
| 4×4 convT(ngf × 4), LReLU | 16×16×(ngf × 4) | 2 |
| 4×4 convT(ngf × 2), LReLU | 32×32×(ngf × 2) | 2 |
| 3×3 convT(3), Tanh | 32×32×3 | 1 |
| Generator Model for CelebA, ngf = 128 | | |
| Input: $z$ | 1×1×100 | |
| 4×4 convT(ngf × 8), LReLU | 4×4×(ngf × 8) | 1 |
| 4×4 convT(ngf × 4), LReLU | 8×8×(ngf × 4) | 2 |
| 4×4 convT(ngf × 2), LReLU | 16×16×(ngf × 2) | 2 |
| 4×4 convT(ngf × 1), LReLU | 32×32×(ngf × 1) | 2 |
| 4×4 convT(3), Tanh | 64×64×3 | 2 |

Table 15: Generator architectures used for BinaryMNIST, CIFAR-10, and CelebA datasets. convT($n$) denotes a transposed convolution with $n$ output feature maps.

| Encoder Model for BinaryMNIST, nef = 16 | | |
|---|---|---|
| Layers | In-Out Size | Stride |
| Input: $x$ | 1×28×28 | |
| 5×5 conv(ngf × 2), LReLU | 14×14×(ngf × 2) | 2 |
| 5×5 conv(ngf × 4), LReLU | 7×7×(ngf × 4) | 2 |
| 5×5 conv(ngf × 8), LReLU | 4×4×(ngf × 8) | 2 |
| Linear | 16 | - |
| Encoder Model for CIFAR-10 (matches code), nef = 100 | | |
| Layers | In-Out Size | Stride |
| Input: $x$ | $nc$×32×32 | |
| 4×4 conv($nef$), LReLU | 16×16×($nef$) | 2 |
| 4×4 conv($nef$×2), LReLU | 8×8×($nef$×2) | 2 |
| 4×4 conv($nef$×4), LReLU | 4×4×($nef$×4) | 2 |
| 4×4 conv($nef$×8), LReLU | 2×2×($nef$×8) | 2 |
| 4×4 conv($nz$) | 1×1×($nz$) | 1 |
| Flatten | $nz$ | - |
| Linear | $nz$ | - |
| Encoder Model for CelebA (matches code), nef = 100 | | |
| Layers | In-Out Size | Stride |
| Input: $x$ | $nc$×64×64 | |
| 4×4 conv($nef$), LReLU | 32×32×($nef$) | 2 |
| 4×4 conv($nef$×2), LReLU | 16×16×($nef$×2) | 2 |
| 4×4 conv($nef$×4), LReLU | 8×8×($nef$×4) | 2 |
| 4×4 conv($nef$×8), LReLU | 4×4×($nef$×8) | 2 |
| 4×4 conv($nef$×16), LReLU | 2×2×($nef$×16) | 2 |
| 4×4 conv(100) | 1×1×(100) | 1 |
| Flatten | 100 | - |
| Linear | 100 | - |

Table 16: Encoder architectures used for BinaryMNIST, CIFAR-10, and CelebA. For CIFAR-10 and CelebA, the table matches the provided implementation: stacked 4 × 4 Conv2d blocks with stride 2 and pad 1, followed by a final 4 × 4 Conv2d to $nz$, flatten, and a linear layer.

### I.5.1   Sampling from the prior EBM

Samples from the EBM prior are obtained using Langevin dynamics with 80 warm-up steps, a step size of $4 \times 10^{-1}$, and thinning set to 1, from which 64 samples are retained.

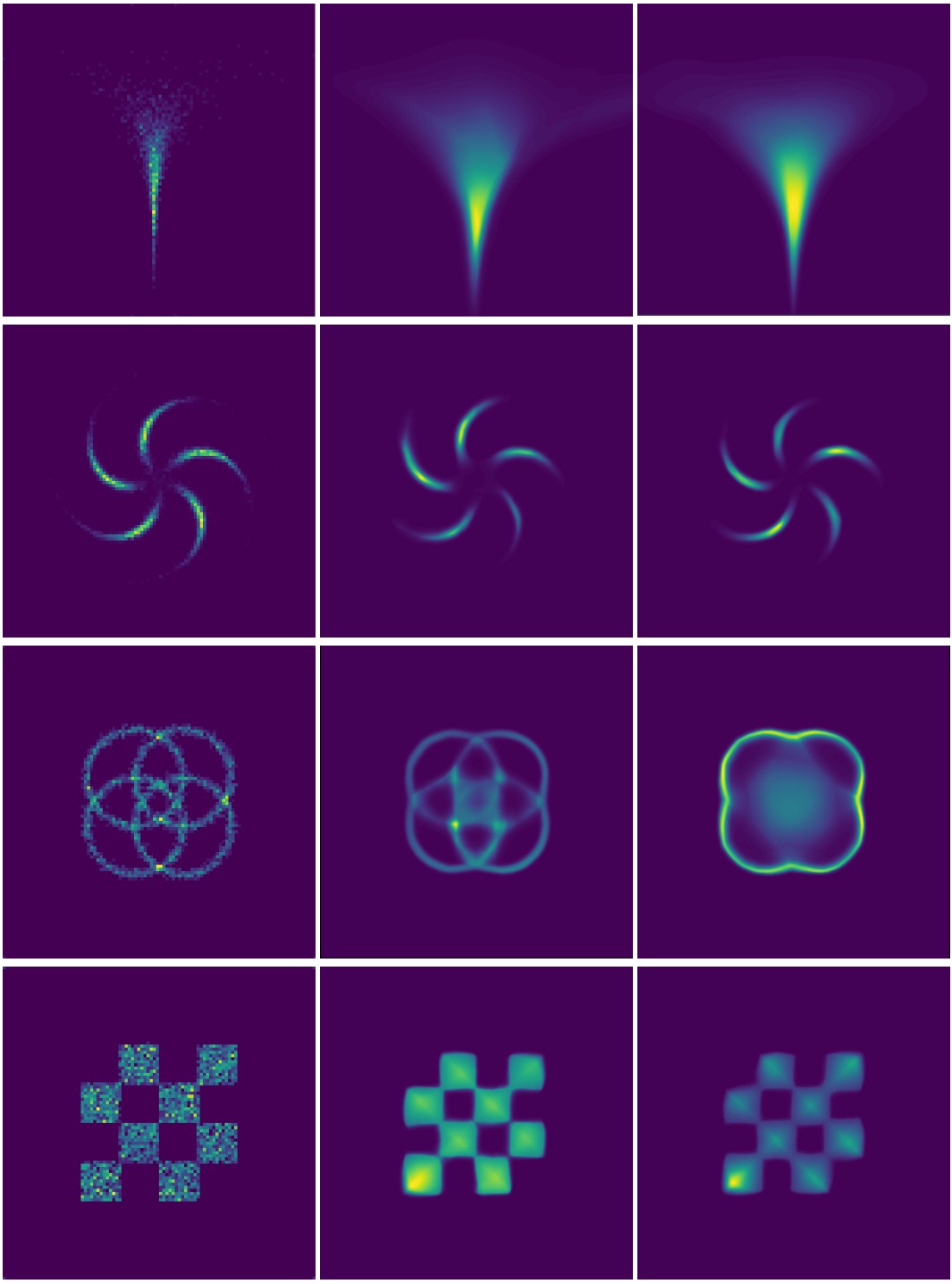

Figure 6: Each row is a dataset, the first column displays samples from the dataset, the second column displays the energy function of an EBM trained with the self normalised log-likelihood (SNL), the third column displays the energy function of an EBM trained with Noise Contrastive Estimation (NCE). We use a standard Gaussian as the base distribution for both training methods. These parameterisations correspond to the first two lines of Table 1.

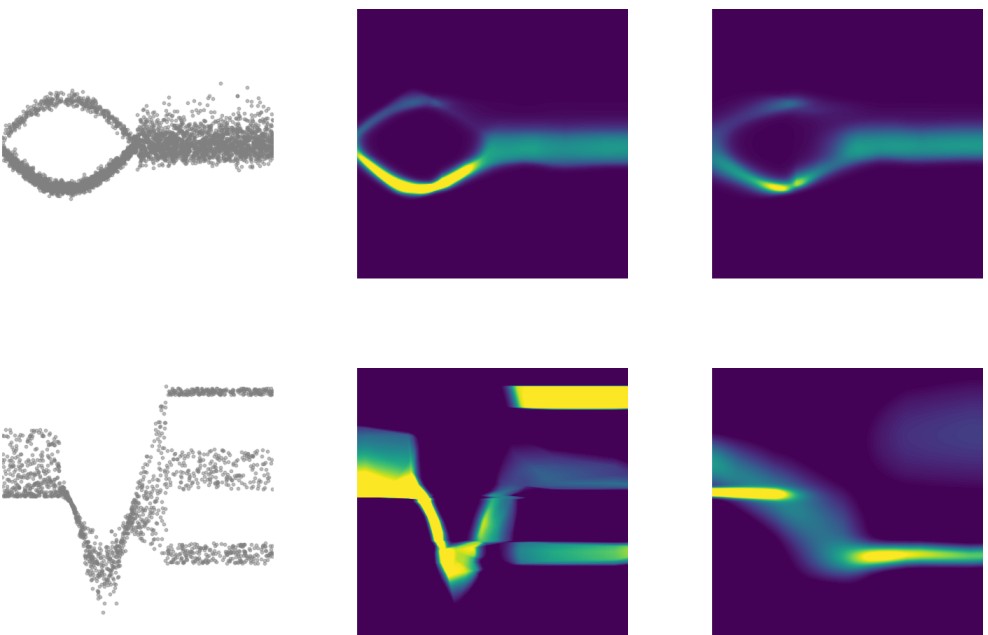

Figure 7: Each row corresponds to a dataset, the first column displays samples from the dataset, the second column displays the energy function of an EBM trained with the self normalised log-likelihood (SNL), the third column displays the energy function of an EBM trained with Noise Contrastive Estimation (NCE). We use a the Mixture density Network (MDN) proposals with $K = 2$ as the proposal for both methods (see Table 3).

## J   Additional results

| | True | ML $\pm$ SE | PL $\pm$ SE | SNL $\pm$ $\sigma$ | SM $\pm$ $\sigma$ |
|---|---|---|---|---|---|
| $\kappa_1$ | 2.00 | 2.66 $\pm$ 0.38 | 2.81 $\pm$ 0.53 | 1.99 $\pm$ 0.28 | 1.67 $\pm$ 0.38 |
| $\kappa_2$ | 3.00 | 2.84 $\pm$ 0.39 | 2.81 $\pm$ 0.44 | 3.06 $\pm$ 0.23 | 2.92 $\pm$ 0.45 |
| $\kappa_3$ | 1.00 | 0.98 $\pm$ 0.21 | 0.93 $\pm$ 0.19 | 0.89 $\pm$ 0.21 | 0.55 $\pm$ 0.20 |
| $\lambda_{12}$ | 2.00 | 2.33 $\pm$ 0.55 | 2.64 $\pm$ 0.80 | 2.31 $\pm$ 0.48 | 2.78 $\pm$ 0.28 |
| $\lambda_{13}$ | 2.00 | 2.58 $\pm$ 0.45 | 2.57 $\pm$ 0.50 | 2.00 $\pm$ 0.31 | 0.16 $\pm$ 0.46 |
| $\lambda_{23}$ | 2.00 | 1.49 $\pm$ 0.48 | 1.17 $\pm$ 0.51 | 1.82 $\pm$ 0.32 | 0.89 $\pm$ 0.55 |
| $\kappa_1$ | 0.50 | 0.82 $\pm$ 0.26 | 0.82 $\pm$ 0.30 | 0.47 $\pm$ 0.19 | 0.82 $\pm$ 0.58 |
| $\kappa_2$ | 0.75 | 0.71 $\pm$ 0.26 | 0.71 $\pm$ 0.26 | 0.81 $\pm$ 0.36 | 0.83 $\pm$ 0.74 |
| $\kappa_3$ | 0.25 | 0.39 $\pm$ 0.26 | 0.40 $\pm$ 0.28 | 0.13 $\pm$ 0.13 | 1.53 $\pm$ 0.86 |
| $\lambda_{12}$ | 2.00 | 2.36 $\pm$ 0.73 | 2.24 $\pm$ 0.68 | 1.87 $\pm$ 0.58 | 2.66 $\pm$ 1.54 |
| $\lambda_{13}$ | 3.00 | 3.27 $\pm$ 0.71 | 3.36 $\pm$ 0.64 | 2.91 $\pm$ 0.63 | 2.95 $\pm$ 1.41 |
| $\lambda_{23}$ | 4.00 | 3.49 $\pm$ 0.70 | 3.53 $\pm$ 0.69 | 4.62 $\pm$ 0.43 | 2.98 $\pm$ 1.77 |
| $\kappa_1$ | 2.00 | 2.65 $\pm$ 0.97 | 2.65 $\pm$ 0.98 | 2.19 $\pm$ 0.92 | 4.44 $\pm$ 4.87 |
| $\kappa_2$ | 2.00 | 1.66 $\pm$ 0.81 | 1.65 $\pm$ 0.85 | 2.04 $\pm$ 0.27 | 5.99 $\pm$ 7.31 |
| $\kappa_3$ | 2.00 | 2.01 $\pm$ 0.85 | 2.02 $\pm$ 0.92 | 1.87 $\pm$ 1.01 | 4.01 $\pm$ 10.0 |
| $\lambda_{12}$ | 20.00 | 36.85 $\pm$ 8.63 | 36.76 $\pm$ 6.99 | 15.6 $\pm$ 8.51 | 8.6 $\pm$ 12.76 |
| $\lambda_{13}$ | 30.00 | 40.01 $\pm$ 8.55 | 40.15 $\pm$ 8.49 | 35.0 $\pm$ 8.11 | 19.26 $\pm$ 16.06 |
| $\lambda_{23}$ | 40.00 | 23.66 $\pm$ 8.61 | 23.64 $\pm$ 7.87 | 40.23 $\pm$ 6.88 | 26.47 $\pm$ 19.97 |
| $\kappa_1$ | 2.00 | 1.84 $\pm$ 0.23 | 1.84 $\pm$ 0.23 | 2.17 $\pm$ 0.14 | 2.20 $\pm$ 0.22 |
| $\kappa_2$ | 2.00 | 1.83 $\pm$ 0.23 | 1.83 $\pm$ 0.23 | 2.10 $\pm$ 0.19 | 2.14 $\pm$ 0.36 |
| $\kappa_3$ | 2.00 | 1.94 $\pm$ 0.24 | 1.94 $\pm$ 0.23 | 2.09 $\pm$ 0.13 | 2.14 $\pm$ 0.21 |
| $\lambda_{12}$ | 0.10 | 0.15 $\pm$ 0.28 | 0.14 $\pm$ 0.28 | -0.12 $\pm$ 0.17 | 2.25 $\pm$ 0.20 |
| $\lambda_{13}$ | 0.10 | 0.17 $\pm$ 0.28 | 0.16 $\pm$ 0.28 | 0.01 $\pm$ 0.19 | 2.29 $\pm$ 0.29 |
| $\lambda_{23}$ | 0.10 | 0.12 $\pm$ 0.28 | 0.12 $\pm$ 0.30 | 0.12 $\pm$ 0.26 | 2.38 $\pm$ 0.20 |

Table 17: Parameters estimates of a multivariate von Mises distribution (Mardia et al., 2008) for 4 different sets of parameters. For this experiment, the location parameters are known and set to 0. The Maximum Likelihood (ML) and Pseudo-Likelihood (PL) results are directly reported from Mardia et al. (2008). In these cases, the uncertainty estimates (SE) are obtained using Jacknife estimators. The score matching estimators are obtained using explicit formulation in Mardia et al. (2016) while the SNL estimators are calculated using gradient descents. We report the average parameters and associated standard deviation over five runs with five different datasets.

| | $\mathcal{L}$ | $||\pi - \hat{\pi}||_2$ | $||\mu - \hat{\mu}||_2$ | $||\Sigma^{-1} - \hat{\Sigma}^{-1}||_F$ |
|---|---|---|---|---|
| EM | $-4.934 \pm 0.419$ | $0.0143 \pm 0.0084$ | $10.87 \pm 11.22$ | $55.66 \pm 35.61$ |
| GMMis | $-4.284 \pm 0.348$ | $0.0043 \pm 0.0029$ | $1.46 \pm 0.84$ | $16.23 \pm 5.74$ |
| SNL | $-4.145 \pm 0.095$ | $0.0066 \pm 0.003$ | $0.87 \pm 0.83$ | $17.89 \pm 4.38$ |

Table 18: Comparative evaluation of the three estimation models (Standard EM Dempster et al. (1977), GMMis Melchior & Goulding (2018) and SNL). We evaluate the likelihood of the model on a non-truncated test dataset to show the quality of the estimated parameters and the norm of the difference with the original parameters. Each entry is reported as mean $\pm$ standard deviation over five generated datasets and runs. Both SNL and GMMis perform on par, outperforming the original EM, but it should be noted that GMMis is much faster than doing gradient descent with SNL.

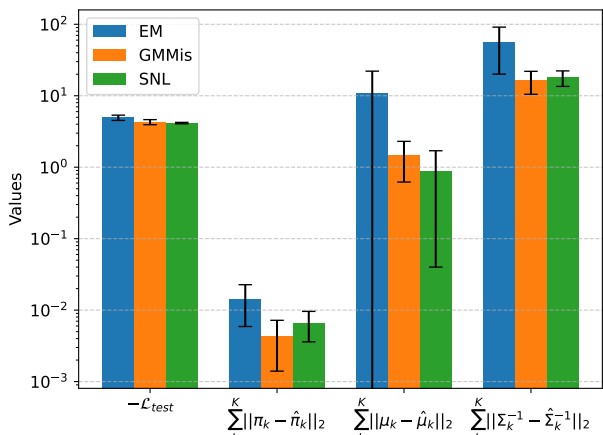

Figure 8: Log-scale bar plot comparing EM, GMMis, and SNL methods for the truncated mixture of Gaussians across four evaluation metrics. Bars represent the absolute mean values, with error bars indicating standard deviation over five different runs and datasets. We report the negative log-likelihood over the untruncated test set and the norm of the difference of estimated parameters with the original parameters.

| Dataset | VAE | SNELBO | short-term MCMC Pang et al. (2020) |
|---|---|---|---|
| CIFAR-10 | 107.57 | 98.36 | **70.15** |
| CelebA | 66.7 | 64.2 | **37.87** |

Table 19: FID of generated samples for CIFAR-10 and CelebA. The results of short-term MCMC are directly reported from Pang et al. (2020). Though using the EBM prior with SNELBO improves slightly the generation, the quality ofthe generation is far from being competitive with latent short term MCMC.

| Models | | | | Datasets | | | |
|---|---|---|---|---|---|---|---|
| | | | | Regression Dataset 1 | | Regression Dataset 2 | |
| Objective | Proposal $q$ | $b_\phi$ | Base Dist | $\ell_{\mathrm{IS}}$ | $\ell_{\mathrm{SNL}}$ | $\ell_{\mathrm{IS}}$ | $\ell_{\mathrm{SNL}}$ |
| NCE | $\mathcal{N}(\mu,\Sigma)$ | None | None | $-0.100$ $(\pm 0.186)$ | $-0.638$ $(\pm 0.168)$ | $-2.416$ $(\pm 0.376)$ | $-3.049$ $(\pm 0.900)$ |
| NCE | $\mathcal{N}(\mu,\Sigma)$ | None | $q$ | $-0.336$ $(\pm 0.468)$ | $-1.567$ $(\pm 0.282)$ | $-2.548$ $(\pm 0.232)$ | $-2.676$ $(\pm 0.169)$ |
| NCE | $\mathcal{N}(\mu,\Sigma)$ | MLP | None | $-0.030$ $(\pm 0.278)$ | $-0.718$ $(\pm 0.256)$ | $-2.592$ $(\pm 0.214)$ | $-3.559$ $(\pm 1.881)$ |
| NCE | $\mathcal{N}(\mu,\Sigma)$ | MLP | $q$ | $-0.644$ $(\pm 0.632)$ | $-1.580$ $(\pm 0.480)$ | $-2.426$ $(\pm 0.257)$ | $-2.586$ $(\pm 0.238)$ |
| NCE | MDN K2 | None | None | $-0.570$ $(\pm 0.209)$ | $-1.275$ $(\pm 0.688)$ | $-2.451$ $(\pm 0.040)$ | $-3.094$ $(\pm 0.515)$ |
| NCE | MDN K2 | MLP | None | $-0.611$ $(\pm 0.154)$ | $-1.492$ $(\pm 0.993)$ | $-2.451$ $(\pm 0.088)$ | $-2.634$ $(\pm 0.084)$ |
| SNL | $\mathcal{N}(\mu,\Sigma)$ | None | None | $0.091$ $(\pm 0.122)$ | $-0.023$ $(\pm 0.071)$ | $-1.597$ $(\pm 0.047)$ | $-1.619$ $(\pm 0.063)$ |
| SNL | $\mathcal{N}(\mu,\Sigma)$ | None | $q$ | $0.065$ $(\pm 0.084)$ | $-0.044$ $(\pm 0.095)$ | $-1.493$ $(\pm 0.039)$ | $-1.503$ $(\pm 0.041)$ |
| SNL | $\mathcal{N}(\mu,\Sigma)$ | MLP | None | $0.164$ $(\pm 0.088)$ | $0.033$ $(\pm 0.077)$ | $-1.813$ $(\pm 0.109)$ | $-1.836$ $(\pm 0.109)$ |
| SNL | $\mathcal{N}(\mu,\Sigma)$ | MLP | $q$ | $0.091$ $(\pm 0.094)$ | $-0.048$ $(\pm 0.030)$ | $\mathbf{-1.468}$ $(\pm 0.014)$ | $\mathbf{-1.477}$ $(\pm 0.016)$ |
| SNL | MDN K2 | None | None | $0.227$ $(\pm 0.058)$ | $0.221$ $(\pm 0.059)$ | $-2.061$ $(\pm 0.145)$ | $-2.070$ $(\pm 0.141)$ |
| SNL | MDN K2 | MLP | None | $\mathbf{0.255}$ $(\pm 0.017)$ | $\mathbf{0.251}$ $(\pm 0.016)$ | $-2.099$ $(\pm 0.250)$ | $-2.170$ $(\pm 0.353)$ |

Table 20: Evaluation of regression EBMs on the 1D toy regression problems with two different objectives and different sets of parameters. Each model is trained for five runs, and we report the mean and standard deviation of the estimated log-likelihood $\ell_{\mathrm{IS}}$ and the self-normalised log-likelihood $\ell_{\mathrm{SNL}}$. Using the SNL as the objective clearly outperforms the NCE.

| Models | | Datasets | | | | | | | |
|---|---|---|---|---|---|---|---|---|---|
| | | Steering Angle | | Cell Count | | UTKFaces | | BIWI | |
| Objective | Proposal | $\ell_{\mathrm{IS}}$ | $\ell_{\mathrm{SNL}}$ | $\ell_{\mathrm{IS}}$ | $\ell_{\mathrm{SNL}}$ | $\ell_{\mathrm{IS}}$ | $\ell_{\mathrm{SNL}}$ | $\ell_{\mathrm{IS}}$ | $\ell_{\mathrm{SNL}}$ |
| NCE | $\mathcal{N}(\mu,\Sigma)$ | $-3.649$ $(\pm 1.224)$ | Unnormalized | $-3.367$ $(\pm 0.399)$ | $-9.675$ $(\pm 0.605)$ | $-3.147$ $(\pm 0.1100)$ | $-8.223$ $(\pm 3.795)$ | $-11.02$ $(\pm 0.576)$ | Unnormalized |
| NCE | MDN-4 | $-4.044$ $(\pm 0.741)$ | $-10.272$ $(\pm 0.742)$ | $-3.856$ $(\pm 0.029)$ | Unnormalized | $-3.876$ $(\pm 0.140)$ | $-12.093$ $(\pm 0.233)$ | $-12.093$ $(\pm 0.155)$ | Unnormalized |
| NCE | MDN-8 | $-4.001$ $(\pm 0.667)$ | Unnormalized | $-3.864$ $(\pm 0.048)$ | Unnormalized | $-4.123$ $(\pm 0.21)$ | $-5.170$ $(\pm 0.955)$ | $-11.998$ $(\pm 0.339)$ | Unnormalized |
| SNL | $\mathcal{N}(\mu,\Sigma)$ | $-2.665$ $(\pm 1.37)$ | $-3.973$ $(\pm 3.15)$ | $-2.701$ $(\pm 0.041)$ | $-2.725$ $(\pm 0.046)$ | $-2.966$ $(\pm 0.057)$ | $-2.991$ $(\pm 0.069)$ | $-10.86$ $(\pm 1.017)$ | $-11.05$ $(\pm 1.141)$ |
| SNL | Uniform | $\mathbf{-1.402}$ $(\pm 0.068)$ | $\mathbf{-1.423}$ $(\pm 0.074)$ | $\mathbf{-2.604}$ $(\pm 0.001)$ | $\mathbf{-2.620}$ $(\pm 0.007)$ | $-2.927$ $(\pm 0.032)$ | $-2.965$ $(\pm 0.019)$ | $-10.44$ $(\pm 0.138)$ | $-10.51$ $(\pm 1.222)$ |
| SNL | MDN-4 | $-1.780$ $(\pm 0.2312)$ | $-1.795$ $(\pm 0.231)$ | $-2.834$ $(\pm 0.041)$ | $-2.846$ $(\pm 0.043)$ | $-2.992$ $(\pm 0.045)$ | $-3.004$ $(\pm 0.075)$ | $-10.08$ $(\pm 0.149)$ | $-10.11$ $(\pm 0.126)$ |
| SNL | MDN-8 | $-1.673$ $(\pm 0.042)$ | $-1.692$ $(\pm 0.046)$ | $-2.801$ $(\pm 0.071)$ | $-2.811$ $(\pm 0.071)$ | $\mathbf{-2.921}$ $(\pm 0.055)$ | $\mathbf{-2.943}$ $(\pm 0.062)$ | $\mathbf{-10.01}$ $(\pm 0.092)$ | $\mathbf{-10.04}$ $(\pm 0.091)$ |

Table 21: Evaluation of EBMs for regression on image regression datasets with two different objectives and different proposals. Each model is trained for five runs and we report the mean and standard deviation of the estimated log-likelihood ($\ell_{IS}$) and estimated self-normalised log-likelihood ($\ell_{SNL}$). When the proposal is MDN, the proposal is learned jointly with the model following Gustafsson et al. (2022).

