# OpenReview forum: "Learning Energy-Based Models by Self-Normalising the Likelihood"
_TMLR — Accepted by TMLR_

### Review · Reviewer_DScY · 2026-01-03

**Summary Of Contributions:**

This paper proposes a new way to train energy-based models (EBMs) with (approximate) maximum likelihood without MCMC. The core idea is to introduce a self-normalised log-likelihood (SNL) objective by using a simple variational identity that linearises the problematic $\log Z_\theta$ term. Concretely, the method augments the EBM with one extra learnable scalar parameter $b$ intended to represent $\log Z_\theta$. The authors show that (i) for fixed $\theta$, maximizing SNL over $b$ recovers the exact log-likelihood, and (ii) jointly maximizing over $(\theta,b)$ recovers the maximum-likelihood estimate of $\theta$ while also estimating the normalisation constant. :contentReference[oaicite:0]{index=0}

A practical consequence is that SNL depends linearly on $Z_\theta$, enabling unbiased stochastic gradient estimates via importance sampling from a proposal distribution $q$, and therefore direct optimisation with SGD/Adam rather than sampling from the model with long Markov chains. The paper also provides theory, including joint concavity for exponential families and an information-theoretic interpretation via a generalised KL divergence.

**Audience:**

Yes

**Audience Explanation:**

I expect a meaningful subset of TMLR’s audience would be interested

**Broader Impact Concerns:**

I don’t see any specific broader-impact red flags in the submission: it’s primarily a methodological contribution (a training objective for EBMs that avoids MCMC by using a self-normalised log-likelihood and proposal/importance sampling)

**Claims And Evidence:**

Yes

**Claims Explanation:**

Overall, the main claims are supported by solid (and mostly clear) evidence, especially on the theoretical side and on the regression benchmarks.

- Theory / objective claims: The paper’s central claims about SNL (i.e., (i) maximizing over the extra scalar parameter recovers the exact log-likelihood for fixed $\theta$, (ii) joint optima correspond to ML optima, (iii) SNL is a lower bound and can be “sandwiched” with an IS upper bound) are stated precisely and backed by formal results and proofs (main text + appendices).
- Optimization tractability claims: The argument that SNL enables unbiased stochastic gradients via importance sampling from a proposal $q$ is explicitly derived (showing unbiased estimators for $\nabla_\theta$ and $\nabla_b$ under the proposal-sampling scheme), which is convincing as written.
- Empirical claims: The experimental section provides multiple settings (directional/truncated, low-dimensional density estimation, regression, and VAE-with-EBM-prior). The strongest and clearest empirical support is for the claim that SNL training improves over NCE in regression/image regression, supported by repeated runs and tables/figures studying the role of the number of proposal samples $M$.
- Limitations / scope: The paper is reasonably transparent that scalability to very high-dimensional settings remains an open issue.

**Requested Changes:**

I do not have any particular requested changes. The submission is clear and technically sound as written; any further edits would be minor polish/clarifications rather than requirements for my recommendation.

---

> ### Author Response · Authors · 2026-01-29
>
> We thank the reviewer for their careful reading and for the thorough and positive assessment of our work. We are grateful for their detailed summary of the contributions and for highlighting both the theoretical grounding and the empirical strengths of the proposed SNL framework. We also appreciate the reviewer's recognition of the current limitations in high-dimensional settings and their overall assessment that the submission is clear, technically sound, and of interest to the TMLR audience.
>
> Based on the constructive feedback from all reviewers, we have made several improvements to the revised manuscript, including additional experiments on CIFAR-10 and CelebA (Table 19), a new appendix clarifying the relationship with KALE (Appendix E), improved figure captions, and more precise claims regarding performance comparisons. We hope these revisions further strengthen the contribution.

---

### Review · Reviewer_oAn8 · 2026-01-09

**Summary Of Contributions:**

This article introduces a self-normalizing likelihood estimation framework to train energy-based models when the normalization constant is intractable. The novel objective function provides a lower bound for the log-likelihood, and it can be optimized using stochastic gradient descent methods. The effectiveness of the approach is validated on various density estimation, parameter estimation and regression tasks. The effect of the dimensionality of an energy-based model is also discussed.

**Audience:**

Yes

**Audience Explanation:**

Although the SNL loss exists already, its combination with idea of importance sampling is interesting, and the results obtained from this study are worth publishing.

**Claims And Evidence:**

No

**Claims Explanation:**

The article is well written. However, It seems to me that the main idea of optimizing the SNL loss already exists in the literature, therefore some claims should be adjusted or clarified to have a better positioning. The evaluation of some numerical results could also be strengthened to make certain results more convincing.

**Requested Changes:**

-	The loss l_SNL in eq. 10 seems to be the same as the KALE loss in Arbel 2021 et al. (eq. 10). Thus, one should clearly mention the results in this article, as l_SNL is the same as Theorem 2.1. The positioning in 2.5 related works should be made more specific about this point.
-	The results in Fig 2. Evaluate the errors of parameter estimation partially, only with respect to the parameter kappa and lambda. It would be more complete if the error on the theta is also reported.
-	The claim from Table 2 seems to be too strong. How do you conclude that the results outperform or are on par with other method? In particular, the MAE MoG provides a larger log-likelihood estimation -15.15 compared you EBM-SNL in Hepmass. This point should be clarified or reduced.
-	Could you clarify the Fig 5, what is the x-axis / y-axis?
-	As Fig 2 in Gustafasson 2022, you could provide a direct 2d plot on the learnt energy to evaluate the results in 5.2.1 for 1d regression.
-	Is there any explanation why the method stops to improve after M=64 samples in Fig 4? Did you look at the lower bound l_SNL instead the upper bould I_IS? How l_SNL evolves with respect to M?

---

> ### Author Response · Authors · 2026-01-29
>
> We thank the reviewer for the careful reading of our manuscript and for the constructive feedback. Below we address each point in detail.
>
> > The loss l_SNL in eq. 10 seems to be the same as the KALE loss in Arbel 2021 et al. (eq. 10). Thus, one should clearly mention the results in this article, as l_SNL is the same as Theorem 2.1. The positioning in 2.5 related works should be made more specific about this point.
>
> We thank the reviewer for pointing us to this important reference, which helped clarify the relationship between SNL and existing variational formulations of the KL divergence. In the revised version, we explicitly discuss this connection and substantially strengthen the positioning of our work.
>
> In particular, we add a new appendix (Appendix E) showing that the SNL objective can be recovered by evaluating the Fenchel (Donsker-Varadhan) dual formulation of the Kullback-Leibler divergence under a structured parametrisation of the dual variable. This makes the relationship with variational KL representations explicit.
>
> While KALE and SNL both rely on variational formulations of the KL divergence, they differ fundamentally in objective and guarantees. KALE is obtained by restricting and regularising the Fenchel dual of the KL, yielding a surrogate divergence between distributions that depends on the choice of function class $\mathcal{H}$ and the regularisation parameter $\alpha$. As a consequence, its minimiser does not, in general, coincide with the maximum-likelihood solution.
>
> In contrast, SNL is derived by applying a variational identity directly to the logarithm of the normalising constant $Z_\theta$, introducing an auxiliary scalar variable $b$ with a direct probabilistic interpretation. At the optimum, $b$ recovers the log-partition function, and the SNL objective satisfies the exact identity
>
> $$\max_b \, \ell_{\mathrm{SNL}}(\theta,b) = \ell(\theta),$$
>
> yielding a one-to-one correspondence between the optima of SNL and those of the true log-likelihood.
>
> Furthermore, KALE is primarily designed as a divergence-based tool for defining gradient flows or particle dynamics, where the function $h$ is used to transport samples from a source distribution $Q$ toward a target distribution $P$. In our setting, the energy-based model itself represents the final target distribution, and the objective is to learn its parameters for density estimation of $P$.
>
> We revise Section 2.5 accordingly to make these distinctions explicit and to properly acknowledge related work.
>
> > The results in Fig 2. Evaluate the errors of parameter estimation partially, only with respect to the parameter kappa and lambda. It would be more complete if the error on the theta is also reported.
>
> Thank you for pointing out this ambiguity. In Figure 2, we only compare parameters for which the estimators differ across methods. The location parameter $\theta$ admits the same estimator for all methods (it corresponds to a simple empirical average over the training set) and was therefore fixed to its true value.
>
> This experimental setup follows the protocol of Mardia et al. (2008), with whom we compare, where only concentration parameters are reported. We have added this clarification to the figure caption in the revised manuscript to avoid confusion.
>
> > The claim from Table 2 seems to be too strong [...]
>
> We agree with the reviewer and thank them for highlighting this issue. We have revised the corresponding text to adopt a more precise and conservative wording. In particular, MAE MoG achieves −15.15 nats on Hepmass compared to our −15.89, and we now explicitly acknowledge this gap rather than claiming parity.
>
> > Clarification of Figure 5.
>
> We have clarified this in the revised manuscript. The $x$-axis corresponds to the regression input ($x$ in Equation 30), and the $y$-axis corresponds to the regressed output ($y$ in Equation 30). This is now stated explicitly in the figure caption.
>
> > As Fig 2 in Gustafasson 2022,[...]
>
> Following this suggestion, we have added a 2D visualisation of the learned energy for the 1D regression task (Figure 7). The new figure shows the learned energy landscape over the $(x,y)$ plane. This complements the quantitative results previously reported.
>
> > Is there any explanation why the method stops to improve after M=64 samples[...]
>
> Our interpretation is that for small $M$, optimisation is dominated by the variance induced by importance sampling, and increasing $M$ significantly reduces this variance. Beyond a certain point, however, importance-sampling noise is no longer the dominant source of error, and further increasing $M$ yields diminishing returns. Other factors such as mini-batch noise, model capacity, or limited expressiveness of the normalisation network $b_\phi$ may then dominate. Similarly, we analysed the evolution of the lower bound $\ell_{\mathrm{SNL}}$ as a function of $M$ and observed the same saturation behaviour as for the upper bound $\ell_{\mathrm{IS}}$.

---

> > ### Comment · Reviewer_oAn8 · 2026-02-09
> > **clarification**
> >
> > Thanks for your replies. I do not understand the first point regarding the KALE loss in Arbel 2021 et al. (eq. 10). I was talking about the following article:
> > M Arbel, L Zhou, and A Gretton. Generalized energy based models. In International Conference on Learning
> > Representations, 2021.
> > whereas in your Appendix E, i did not see this discussion.
> > Could you check if their Section 3.2, eq. 10 is equivalent to yours?

---

> > > ### Author Response · Authors · 2026-02-12
> > > **Clarification**
> > >
> > > We apologise for the confusion—our initial response addressed KALE Flow (Glaser et al., 2021) rather than the Generalized Energy Based Models (GEBM) paper (Arbel, Zhou, and Gretton, ICLR 2021).
> > >
> > > We confirm that Eq. (10) in GEBM is very similar to our SNL objective. Both are related to the Donsker-Varadhan representation of the KL divergence. **The key difference lies in the role of the base measure:**
> > >
> > > - **In SNL**, using equation (10) is not necessarily feasible as the energy may not be integrable with respect to the Lebesgue measure. Instead, we use an additional base measure $d(x)$ (sec 2.4) that is *fixed* and defines the probabilistic model $p_\theta(x) \propto \exp(-E_\theta(x)) d(x)$. Separately, a proposal $q(x)$ is used for importance sampling to estimate the partition function. These two distributions can be chosen independently—the population-level objective is invariant to the choice of $q$.
> > >
> > > - **In GEBM**, the base measure $\mathbb{B}$ is *learnable* (e.g., a GAN generator), and samples for estimating the partition function are drawn directly from $\mathbb{B}$. There is no separation between base and proposal: both are the same learned distribution, and both change during training. In the special case where $\mathbb{B}$ is fixed, the GEBM objective reduces to a particular case of SNL $d(x)$ as in which the base is equal to the proposal.
> > >
> > >
> > > Beyond the base measure question, the main practical difference between GEBM and SNL is that SNL allow the user to choose a proposal that differs from the base.
> > >
> > >
> > > We have revised Section 2.5 (Related Works) and rewritten the appendix to make these distinctions explicit. We kept the discussion of KALE Flow in Appendix E, as it is also related to SNL. We hope this clarifies the relationship between these methods.

---

### Review · Reviewer_sJRH · 2026-01-17

**Summary Of Contributions:**

This manuscript presents a novel self-normalized log-likelihood strategy designed to improve likelihood estimation. The core contribution is the incorporation of additional learnable parameters to explicitly model the normalization term, leading to significant performance gains. Extensive experiments across diverse tasks convincingly demonstrate the method's efficacy.

**Audience:**

Yes

**Audience Explanation:**

The computation of normalization terms has long been a notorious challenge in energy-based models (EBMs). The strategy employed in this paper—approximating these terms via learnable parameters is highly valuable and offers a practical solution. Overall, I find the research meaningful and the findings significant to the field.

**Claims And Evidence:**

Yes

**Claims Explanation:**

The authors provide rigorous proofs for all theorems and statements in the appendix. The inclusion of comprehensive derivations and clear algorithmic descriptions significantly enhances the theoretical soundness and reproducibility of the work.

**Requested Changes:**

Although the authors acknowledge the method's limitations in high-dimensional scenarios, I strongly recommend applying it to the CIFAR-10 dataset and reporting the FID metric. Even if the results are not state-of-the-art, analyzing the performance on this dataset could provide valuable insights for future extensions of the method to higher-dimensional spaces.

---

> ### Author Response · Authors · 2026-01-29
>
> We thank the reviewer for the positive assessment of our work and for highlighting its relevance and practical significance for energy-based models.
>
> > Although the authors acknowledge the method's limitations in high-dimensional scenarios, I strongly recommend applying it to the CIFAR-10 dataset and reporting the FID metric. Even if the results are not state-of-the-art, analysing the performance on this dataset could provide valuable insights for future extensions of the method to higher-dimensional spaces
>
> Following this suggestion, we have extended our experimental evaluation to include CIFAR-10 and CelebA, and now report the Fréchet Inception Distance (FID) for generation results in Table 19. We use a similar architecture to the Latent-Space Short-Term MCMC Prior VAE model [1], which is described in Appendix I.5, and include their reported results as a reference point for comparison. Sampling from the EBM prior is performed using Langevin dynamics.
>
> The results (FID of 98.36 on CIFAR-10 and 64.2 on CelebA) show that incorporating SNELBO leads to a consistent but modest improvement in generative quality compared to a vanilla VAE (107.57 and 66.7, respectively). However, the performance remains below that of the approach proposed in [1] (70.15 and 37.87). We believe these findings provide useful insight into the behaviour of SNL-based training objectives in higher-dimensional settings. In particular, while our method does not yet reach state-of-the-art generative performance, it has the advantage of providing a valid lower bound on the likelihood of the full model, which is not available in [1].
>
> [1] Pang, B., Han, T., Nijkamp, E., Zhu, S. C., & Wu, Y. N. (2020). Learning latent space energy-based prior model. Advances in Neural Information Processing Systems, 33, 21994-22008.

---

> > ### Comment · Reviewer_sJRH · 2026-01-30
> >
> > Thank you for your response. I have no further questions. I find this an interesting work and recommend accepting it.

---

### Decision · Action_Editor_tvyn · 2026-02-28

**Recommendation:** Accept with minor revision

**Additional Comments:**

In addition to the comment in the Would at least some individuals in TMLR's audience be interested in knowing the findings of this paper? section, which I encourage the authors to explore why the proposed method is not comparable to the current SoTA and add more experiments/explanations in the revision, one reviewer also requests to rephrase the motivation of the loss with respect to the literature, e.g., the GEBM paper. Please carefully revision and the paper and add necessary clarifications in the final revision.

**Audience:**

Yes

**Audience Explanation:**

Dealing with the normalization constraint in EBM learning is a longstanding problem. Researchers at generative models would find the method worth exploring. However, given that the proposed method is only supported by experiments in low-dimension setting, I don't expect it to attract much interest in the community.

**Claims And Evidence:**

Yes

**Claims Explanation:**

This paper proposes a new method that allows the normalization constant in the EBM model to be updated in the learning. The claims are supported by detailed derivations, theory and some experiments on the low-dimensional setting. The reviewers are generally satisfied with the paper and the rebuttal and are supported.